# Architecture and regulation of a GDNF-GFRα1 synaptic adhesion assembly

F. M. Houghton ●[1], S. E. Adams ●[1,5], A. S. Ríos[2], L. Masino ●[3], A. G. Purkiss ●[3], D. C. Briggs ●[1], F. Ledda ●[2] & N. Q. McDonald ●[1,4] ✉

Glial-cell line derived neurotrophic factor (GDNF) bound to its co-receptor GFRα1 stimulates the RET receptor tyrosine kinase, promoting neuronal survival and neuroprotection. The GDNF-GFRα1 complex also supports synaptic cell adhesion independently of RET. Here, we describe the structure of a decameric GDNF-GFRα1 assembly determined by crystallography and electron microscopy, revealing two GFRα1 pentamers bridged by five GDNF dimers. We reconstituted the assembly between adhering liposomes and used cryo-electron tomography to visualize how the complex fulfils its membrane adhesion function. The GFRα1:GFRα1 pentameric interface was further validated both in vitro by native PAGE and in cellulo by cell-clustering and dendritic spine assays. Finally, we provide biochemical and cell-based evidence that RET and heparan sulfate cooperate to prevent assembly of the adhesion complex by competing for the adhesion interface. Our results provide a mechanistic framework to understand GDNF-driven cell adhesion, its relationship to trophic signalling, and the central role played by GFRα1.

Neuronal synapses are asymmetric junctions that form in a highly organised and dynamic process through physical contact with target tissues or neurons[1]. Membrane-bound synaptic adhesion molecules (SAMs) influence neuronal targeting and bidirectional synapse formation by promoting *trans*-synaptic interactions that are either homophilic (protocadherins, synCAMs) or heterophilic (neurexin-neuroligin)[2,3]. A third group of *trans*-synaptic adhesion molecules are ligand-dependent cell adhesion molecules (LiCAMs), which require that their cognate ligands couple the adhesion receptors in trans across opposing membranes[4,5]. Such ligands include cerebellin (which binds presynaptic neurexin and postsynaptic GluD[6]), neurotrophin-3 (which binds to postsynaptic TrkC[7] that bridges presynaptic PTPσ[7]), and glial cell-derived neurotrophic factor (GDNF), the focus of this study, which binds both pre and postsynaptic GFRα1[4].

GDNF is the prototypic member of the GDNF family of dimeric ligands (GFL) that includes neurturin (NRTN)[8], artemin (ARTN)[9],

persephin (PSPN)[10] and a remote homologue GDF15[11–14]. GFLs are able to bind a cognate GFRα co-receptor with high specificity to form five membrane-linked complexes; GDNF-GFRα1[15], NTN-GFRα2[16], ARTN-GFRα3[17], PSPN-GFRα4[18] and GDF15-GFRAL[11–14]. GFRα family members consist of either two or three extracellular cysteine-rich helical domains (D1-3) followed by a flexible C-tail (CT) and either a glyco-phosphatidylinositol (GPI)-anchor or a transmembrane domain (GFRAL)[11,19,20]. Each GFL-GFRα complex has a 2:2 stoichiometry (GFL$_2$-GFRα$_2$) that adopts a U-shaped structure with variable hinge angles[11,20–22]. GDNF$_2$-GFRα1$_2$ acts at the cell membrane by signalling through the RET receptor tyrosine kinase to drive neuronal differentiation, migration and survival within the developing nervous system and kidney[16,23–26].

Therapeutic interest in GDNF has stemmed from its known neuroprotective action on midbrain dopaminergic neurons both in vitro and in vivo[23–25,27]. This neurotrophic behaviour is driven by engagement

[1]Signalling and Structural Biology laboratory, The Francis Crick Institute, 1 Midland Road, London NW1 1AT, UK. [2]Fundación Instituto Leloir, Instituto de Investigaciones Bioquímicas de Buenos Aires, Av. Patricias Argentinas 435, C1405BWE Buenos Aires, Argentina. [3]Structural Biology Science and Technology Platform, The Francis Crick Institute, 1 Midland Road, London NW1 1AT, UK. [4]Institute of Structural and Molecular Biology, Department of Biological Sciences, Birkbeck College, Malet Street, London WC1E 7HX, UK. [5]Present address: Vertex Pharmaceuticals, 86-88 Jubilee Avenue, Milton Park, Abingdon, Oxfordshire OX14 4RW, UK. ✉e-mail: Neil.McDonald@crick.ac.uk

and activation of RET by the GDNF-GFRα1 2:2 complex[26,28–30]. The bipartite GDNF-GFRα1 complex binds two copies of the RET extracellular module (RET$^{ECM}$) to form 2:2:2 hexameric tripartite assemblies *in cis* (same membrane), thereby promoting RET activation through homodimerization and intracellular autophosphorylation[31–33]. The basis for this interaction has been visualized in single particle cryo-EM structures of GDNF$_2$-GFRα1$_2$-RET$^{ECM}_2$ that revealed a two site-recognition of GDNF-GFRα1 by RET$^{ECM}$, driven by both ligand and co-receptor interactions[31,32].

Further complexity is apparent in RET signalling as GDNF-induced neurite outgrowth of dorsal root ganglia cultures and motility of Madin-Darby canine kidney (MDCK) cells through the RET receptor is known to require the presence of cell-surface heparan sulphate (HS) glycosaminoglycans (GAGs)[34,35]. High-affinity GAG binding sites have been identified in both GDNF and GFRα1[19,36], although the functional relevance of GAG binding to each component has yet to be determined.

Evidence of RET-independent functions of GDNF-GFRα1 has been reported[37,38], consistent with the known widespread expression of GFRα family co-receptors in the brain[39,40]. GDNF-GFRα1 can also signal through the neuronal cell adhesion molecule, NCAM, as an alternative receptor for GDNF-GFRα1 in neurons that lack RET[41]. Further, GFRα1 can function in its own right as a LiCAM in the presence of GDNF to facilitate synapse differentiation[4,5]. GFRα1 is localised on both synaptic membranes and in hippocampal neurons (a region known to lack RET expression), promotes both pre- and postsynaptic differentiation in the presence of GDNF[4,42]. This LiCAM function of GDNF-GFRα1 has yet to be explained mechanistically, however.

Here, we describe the architecture of a decameric GDNF-adhesion assembly and reconstitute the assembly in the act of driving liposome adhesion. We employ a variety of structural, biochemical and cell-based assays to validate the complex and its interfaces. Two competing regulatory inputs are identified from RET and HS that disrupt adhesion assembly and instead promote trophic support. Our results explain the basis for GDNF-driven adhesion and its relationship to trophic signalling.

## Results

### Identification of a GDNF-GFRα1 decameric assembly

As part of a previous structural study on the zebrafish GDNF (zGDNF)-GFRα1 (zGFRα1) complex, we identified a monoclinic crystal form that had much larger unit cell constants than expected. We considered whether this form could contain a multivalent assembly of GDNF-GFRα1 that could account for its adhesion function. The monoclinic crystals were grown using recombinant zGFRα1 residues 20-353, (defined as zGFRα1$^{D1-D3}$, lacking the GPI-anchor) in complex with zGDNF residues 134–235 (mature form of GDNF, defined hereafter as zGDNF$^{mat}$). Diffraction data were recorded from these crystals to 2.65 Å and the structure was determined by molecular replacement using the GDNF-GFRα1 1:1 complex (PDB: 3FUB) structure as a search model (Fig. 1a, Supplementary Fig. 1a). Ten unique search solutions were found. When transformed into a common asymmetric unit, the structure appeared to constitute an unexpected "barrel"-shaped decameric assembly. The barrel-shaped assembly has approximate D5 dihedral symmetry, consistent with the calculated self-rotation function from the diffraction data (Supplementary Fig. 1b). The final refined model contained 23,835 non-hydrogen atoms and was refined with an R$_{free}$ of 28.0% and R factor of 23.8% (Table 1).

Within the decameric assembly, the five 'U'-shaped GDNF$_2$-GFRα1$_2$ sub-complexes associate with one another about the five-fold axis through GFRα1:GFRα1 interactions from the tips of the 'U'. This arrangement generates two GFRα1 pentamer rings (green in Fig. 1b) that form the base and lid of the barrel, connected by five covalent GDNF dimer 'staves' (yellow/orange in Fig. 1b). The barrel molecular dimensions are 138 Å in length with a diameter of 118 Å. Each covalent

zGDNF dimer contains a molecular dyad perpendicular to the molecular five-fold axis, and a further molecular dyad is evident that relates pairs of zGDNF$^{mat}_2$-zGFRα1$^{D2-D3}_2$ dimers to each another. The base pentamer is rotated by approximately 36° about the shared five-fold axis relative to the lid pentamer (Supplementary Fig. 1c). Electron density can be seen extending from the specific N-glycosylation site of all zGDNF protomers and a total of 8 N-linked glycans for the 10 zGFRα1 protomers are well resolved within the barrel structure (Supplementary Fig. 1d). Missing from the structure is the zGFRα1 D1 domain, which we found to be removed by time-dependent proteolytic clipping during crystallisation (Supplementary Fig. 1e). The ordered portion of zGFRα1 in the assembly is defined as zGFRα1$^{D2-D3}$.

Each zGFRα1 protomer within the barrel uses its D2 domain to engage the β-finger elements of a zGDNF dimer as previously reported[32]. The ten GFRα1$^{D2-D3}$ protomers within the barrel are almost identical to those observed in the RET ternary complex with GDNF (0.776 Å for 190 Cα atoms)[32] (Supplementary Fig. 1f, g). Unique to the barrel however, is a homophilic GFRα1:GFRα1 interface generated by the non-crystallographic five-fold rotational symmetry (Fig. 1c). To form this interface, helices α12 and α15 from the D3 domain of one GFRα1 molecule (grey in Fig. 1d) dock into a groove formed by helix α9, helix α14 and helix α16 in the adjacent GFRα1 protomer (green in Fig. 1d). The interface covers a total surface area of 622.3–736.4 Å$^2$ as determined by PDBe PISA[43] and has a hydrophobic character. The interface includes L247/V251 from helix α12, L289/L290 from helix α14 and F348 from helix α16 (Fig. 1d), together with several hydrogen bonds and charge interactions at the interface periphery. For example, K254 from helix α12 forms a predicted salt bridge with D287 from helix α14, and the side-chain of E242 from the loop preceding helix α12 forms a predicted salt bridge with K206 from helix α9.

### In vitro decamer reconstitution and validation of homophilic GFRα1 interaction

We hypothesised that the unexpected decameric assembly seen in our crystals may relate to the reported *trans*-adhesion function of GDNF-GFRα1 complexes, and that it assembles by oligomerisation of individual GDNF$_2$-GFRα1$_2$ complexes (Fig. 1e). We further speculated that the structure might reflect a soluble surrogate for an adhesion complex, despite lacking the GPI anchor of each GFRα1 subunit. To ask whether the complex also forms in solution, we used native-PAGE to monitor decamer assembly in vitro. When we expressed a truncated form of zGFRα1 corresponding to that seen in the crystal structure−lacking the D1 domain (defined hereafter as zGFRα1$^{D2-D3+}$) – a higher molecular weight band at ~700 kDa rapidly appeared in the presence of zGDNF$^{mat}$ (Fig. 2a, right). The apparent molecular weight in native PAGE of the major band seen using zGFRα1$^{D2-D3+}$ is consistent with the calculated molecular weight of the crystallographically-observed barrel complex. It suggests a stoichiometry of 10 copies of zGDNF$^{mat}$ plus 10 copies of GFRα1$^{D2-D3+}$. Interestingly, a similar assembly also formed – but much more slowly – when D1 of zGFRα1 was included in the expressed protein construct (in zGFRα1$^{20–368}$), defined hereafter as zGFRα1$^{D1-D3+}$ (Supplementary Fig. 2a). By assessing zGFRα1 structural integrity at each time point using SDS-PAGE, we found that assembly of this higher molecular weight species occurs only following clipping of the zGFRα1 D1 domain (Supplementary Fig. 2a). These data imply that the rate limiting step for barrel assembly in vitro is loss of the D1 domain. Presence of the D1 domain appears to impede formation of the decameric complex in vitro, and zGFRα1$^{D2-D3+}$ appears to be a portion of zGFRα1 that is sufficient to drive GDNF binding and decameric complex formation in vitro. Evidence of additional low molecular weight oligomers by native-PAGE argues either that an equilibrium between distinct assembly intermediates exists or that partially-formed complexes are trapped (Fig. 2a). The precise composition of these intermediate states is unclear, but the array likely corresponds to the addition of each 2:2 complex into the assembly, i.e., 1 × 2:2 up to 4 × 2:2

complexes. Upon reaching 5 ×2:2 complexes with a molecular mass of ~700 kDa, no further assembly products accumulate, consistent with formation of the closed decameric assembly seen crystallographically.

To validate the crystallographic zGDNF-zGFRα1 barrel assembly interfaces using native PAGE, we engineered a double mutant (K254E-L290E) in zGFRα1$^{D2-D3+}$ to target both unique interfaces used by each zGFRα1 subunit within each pentameric ring (Fig. 2b). His-tagged zGFRα1$^{D2-D3+}$-K254E-L290E in complex with an untagged zGDNF$^{mat}$ was

purified by Ni$^{2+}$ sepharose affinity purification and we confirmed by SDS-PAGE that GDNF binding was not perturbed by these mutations (Fig. 2a). Kinetic analysis by native-PAGE revealed that zGFRα1$^{D2-D3+}$-K254E-L290E exhibited a near complete loss of the higher molecular weight species at ~700 kDa corresponding to a complete barrel assembly (Fig. 2a). This provides in vitro evidence that the GFRα1: GFRα1 interface contributes to the formation of the pentamer in the decameric complex.

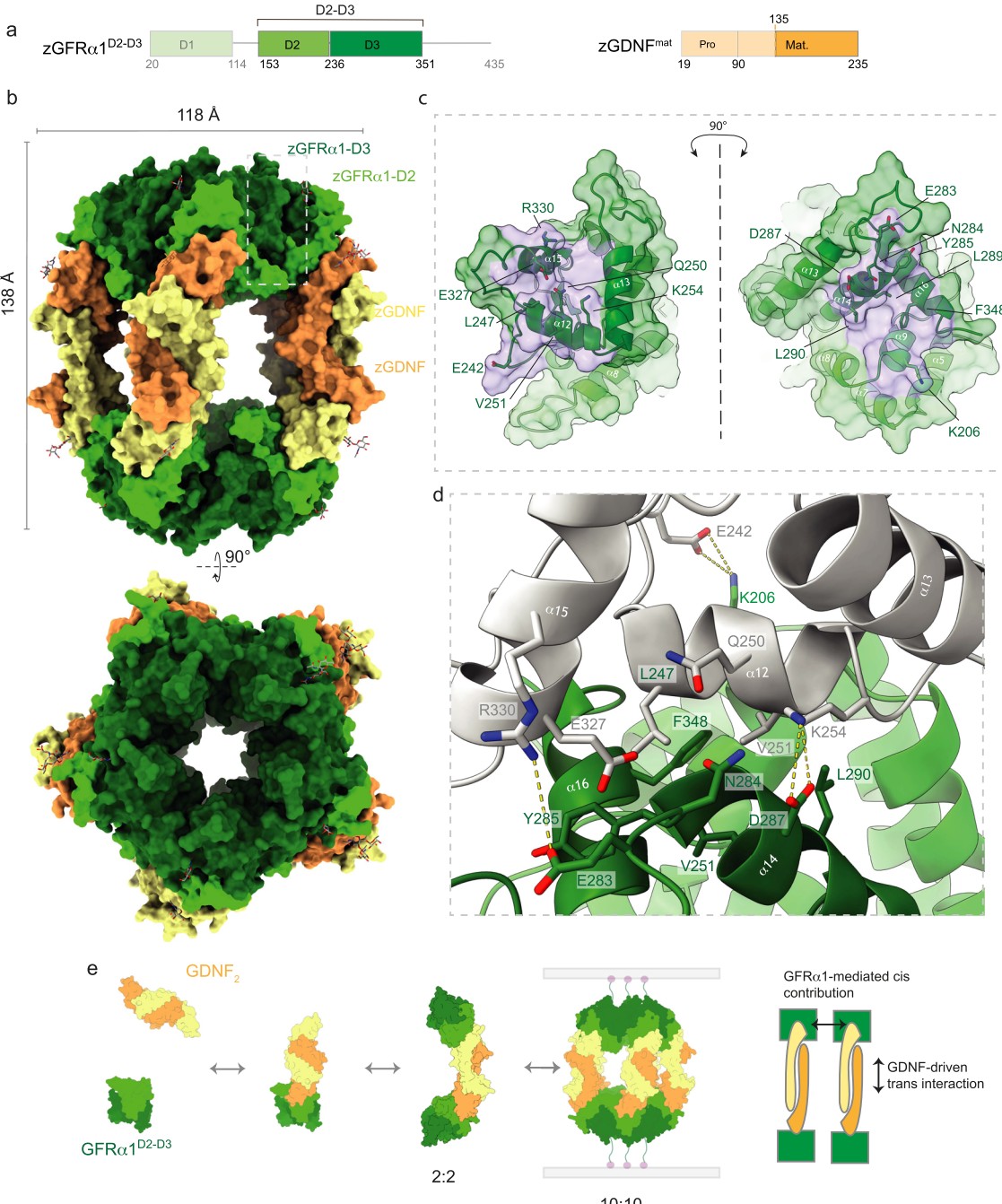

**Fig. 1 | Crystallographic evidence for a decameric assembly of zGDNF-zGFRα1$^{D2-D3}$. a** Domain organisation for zGFRα1$^{D2-D3}$ and zGDNF$^{mat}$ with discrete colours for individual domains. **b** Orthogonal views of the zGFRα1$^{D2-D3}$-zGDNF$^{mat}$ assembly crystal structure. Domains are coloured according to (**a**), zGFRα1-D2, light green, zGFRα1-D3 dark green, GDNF dimer protomers yellow and orange. N-glycans attached to zGDNF and zGFRα1-D3 are shown as sticks. Bottom view projects directly down the five-fold rotational symmetry axis. **c** zGFRα1 interface between each subunit of the pentameric ring, opened to highlight secondary

structure elements and interacting residues at the site of contact. Main chain atoms are shown as a cartoon and interaction residues as sticks. A transparent surface (green) and interaction surface (purple) are also shown. **d** Close up view of the zGFRα1 pentameric interface with key interacting residues shown as sticks with hydrogen-bonds displayed as dashed lines. **e** Schematic model for zGDNF$^{mat}$-zGFRα1$^{D2-D3}$ decameric complex assembly in vitro. The model emphasises a 2:2 zGDNF$^{mat}$-zGFRα1$^{D2-D3}$ complex intermediate that multimerises through GFRα1 homophilic interactions.

## Table 1 | X-ray data collection and refinement statistics

|  | GDNF$_{10}$ – GFRα1$_{10}$ (PDB code: 8OS6) |
|---|---|
| Resolution range | 85.02–2.65 (2.70–2.65) |
| Space group | P2$_1$ |
| Unit cell (a,b,c, α,β,γ) | 114.1 170.0 130.8 90 96.2 90 |
| Total reflections | 1898819 (80645) |
| Unique reflections | 140996 (6703) |
| Multiplicity | 13.5 (12.0) |
| Completeness (%) | 99.1 (95.5) |
| Mean I/sigma(I) | 4.75 (0.25) |
| Wilson B-factor | 57.69 |
| R-pim | 0.1078 (4.085) |
| CC1/2 | 0.990 (0.281) |
| CC* | 0.996 (0.691) |
| Reflections used in refinement | 125,175 (4121) |
| Reflections used for R-free | 6139 (195) |
| R-work | 0.2383 (0.5179) |
| R-free | 0.2797 (0.5074) |
| CC(work) | 0.929 (0.627) |
| CC(free) | 0.929 (0.567) |
| Number of non-hydrogen atoms | 23835 |
| macromolecules | 23416 |
| ligands | 689 |
| solvent | 19 |
| Protein residues | 3001 |
| RMS(bonds) | 0.002 |
| RMS(angles) | 0.47 |
| Ramachandran favoured (%) | 96.48 |
| Ramachandran allowed (%) | 3.49 |
| Ramachandran outliers (%) | 0.03 |
| Rotamer outliers (%) | 3.50 |
| Clashscore | 4.80 |
| Average B-factor | 82.70 |
| macromolecules | 82.57 |
| ligands | 90.75 |
| solvent | 70.04 |
| Number of TLS groups | 108 |

Statistics for the highest-resolution shell are shown in parentheses.

To confirm that the ~700 kDa native-PAGE band is comprised of the decameric zGDNF-zGFRα1 barrel, recombinant zGDNF$^{mat}$-zGFRα1$^{D2-D3+}$ complexes were crosslinked in batch with glutaraldehyde and further purified by size exclusion chromatography (Supplementary Fig. 2b, c). The fraction corresponding to the ~700 kDa band, as assessed by SDS-PAGE and native-PAGE (Supplementary Fig. 2c, g), was imaged by negative-stain electron microscopy (NS-EM). Particles with apparent five-fold rotational symmetry were seen in the raw NS-EM micrographs and 2D class averages (Fig. 2c, d Supplementary Fig. 2d, e). A low-resolution NS-EM single particle reconstruction of the reconstituted zGDNF$^{mat}$$_{10}$-zGFRα1$^{D2-D3+}$$_{10}$ complex was determined at a resolution of 30 Å as estimated by gold-standard Fourier shell correlation (Fig. 2e, Supplementary Fig. 2f, h).

The overall shape of the 3D reconstruction closely resembles the architecture of the barrel crystal structure, with similar approximate D5 dihedral symmetry (Fig. 2e). Some notable differences between the EM and crystal structure can be observed, however. The EM analysis suggests that the two pentameric rings are stacked in an approximately linear manner above each other, without the relative 36°

rotation about the five-fold symmetry axis seen in the crystal structure (Supplementary Fig. 1c). Moreover, the GDNF dimer "staves" adopt a more acute, upright position in the EM reconstruction than in the crystal structure, with distinct GDNF homodimer bend angles (Fig. 2e). These differences suggest a significant degree of conformational plasticity in the zGDNF-zGFRα1 co-receptor-ligand complex, leading to deviations from exact 522 symmetry. These data validate formation of the higher molecular weight species in solution and confirm that it represents the barrel assembly seen in our crystals.

### *Trans*-complex reconstitution on liposomes is sensitive to *cis* interface mutations and RET$^{ECM}$

To probe the impact of GFRα1 pentamer interface mutations on possible GDNF-adhesion function in a membrane context, we next attempted to reconstitute GDNF-dependent GFRα1-mediated *trans*-adhesion using a liposome-based assay. Briefly, unilamellar extruded liposomes containing a mixture of DOPC and DGS-NTA lipids were coated with C-terminally His$_6$-tagged zGFRα1$^{D2-D3+}$ to mimic membrane attachment of zGFRα1 via its GPI modification. Liposome adhesion by zGFRα1 was monitored by time-course measurements of the increase in light scattering at 650 nm (OD650) (Fig. 3a). A robust increase in light scattering was observed upon GDNF addition, demonstrating that liposome clustering is GDNF-driven and does not require D1 of GFRα1—consistent with the in vitro assembly assays described earlier (Figs. 1b, 2a). As a control, we also tested the effect of a zGFRα1$^{D2-D3+}$ mutation (R170E) that disrupts GDNF binding (Supplementary Fig. 4c), for which only a minimal increase in OD650 was observed even in the presence of zGDNF$^{mat}$ (Fig. 3a, e). The double *cis* interface mutant, zGFRα1$^{D2-D3+}$-K254E L290E also led to reduced liposome adhesion capacity upon zGDNF$^{mat}$ stimulation, further validating cooperative behaviour through a pentameric *cis* interface contribution (Fig. 3a, e). The greater impact on the adhesion capacity of GFRα1 of the single *trans* mutant compared to the double *cis* mutant suggests that *trans*-synaptic complex formation is driven by stronger GDNF-mediated interactions in trans and by weaker pentameric interactions *in cis* between GFRα1 subunits.

To confirm that the increased liposome adhesion arose from decameric GDNF-GFRα1 complexes that bridge two liposome membranes, we next imaged zGDNF$^{mat}$-zGFRα1$^{D2-D3+}$-mediated liposome aggregates by cryo-electron tomography (cryo-ET). Tomographic reconstructions revealed clear evidence of protein density bridging across two membranes at sites of liposome contact (Fig. 3b, Supplementary Fig. 3a, b). Intermembrane assemblies have molecular dimensions consistent with the measurements of the decameric complex from the crystal and EM structures. Some views show parallel membrane arrangements of the liposome membranes, consistent with deformation due to adhesion complexes. In other views of stacked liposome pairs formation of an object with five-fold symmetry could be discerned (Fig. 3b, Supplementary Fig. 3b). These objects also displayed a distinctive bilobal feature similar to that observed in the 2D class averages of the NS-EM envelope of zGDNF$^{mat}$$_{10}$-zGFRα1$^{D2-D3+}$$_{10}$ (Fig. 2d)—corresponding to a side-view in which two U-shaped GDNF$_2$-GFRα1$_2$ assemblies on each side of the decameric complex are aligned. To confirm that the objects observed at sites of liposome contact do correspond to decameric GDNF-GFRα1 assemblies, we performed subtomogram averaging of densities picked between liposomes. This analysis yielded a 22 Å resolution map of the objects (Fig. 3c, Supplementary Fig 3c, d). The reconstructions showed clearly recognizable features of two pentameric rings with density for five bridging staves, into which the crystallographic structure could be fit readily, with a correlation coefficient of 0.96. This close correspondence argues that decameric complexes are indeed the bridging component in our reconstituted GDNF-GFRα1 liposome adhesion assay. Furthermore it demonstrates that the assembly can be formed by an un-crosslinked sample and from purified individual components. Thus our cryo-ET

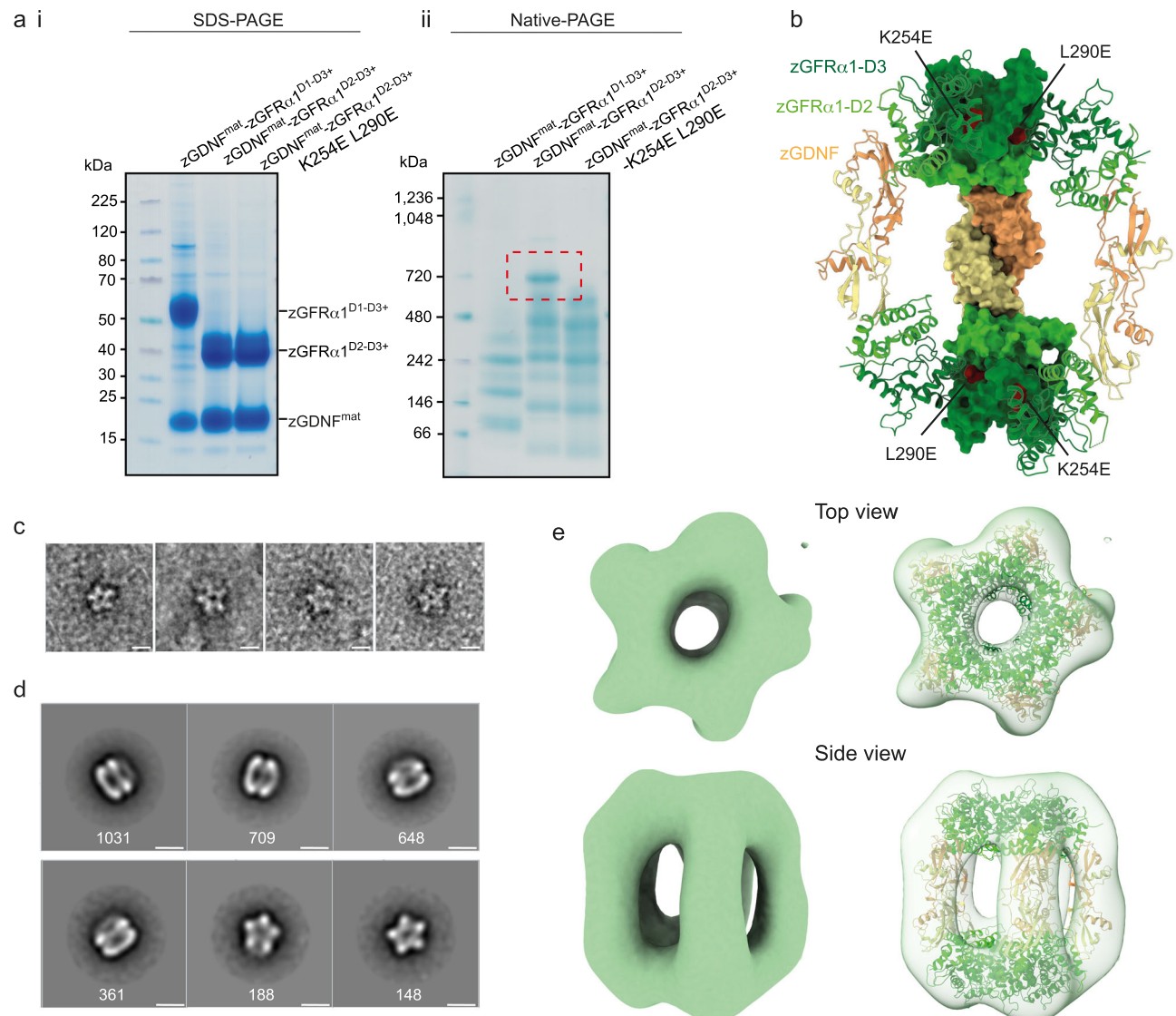

**Fig. 2 | Validation of a zGDNF-zGFRα1<sup>D2-D3+</sup> decameric complex in vitro.**
**a** Reducing SDS-PAGE gel (i) and the corresponding native-PAGE gel (ii) of purified zGFRα1 proteins (as indicated) in complex with untagged zGDNF<sup>mat</sup>. Native-PAGE gel shows the ~700 kDa species of zGDNF<sup>mat</sup>-zGFRα1<sup>D2-D3+</sup> (red dashed box) that is disrupted by the *cis* interface double mutant, zGFRα1<sup>D2-D3+</sup>-K254E L290E, when co-expressed with zGDNF<sup>mat</sup>. SDS-PAGE gel confirms the identity of individual components and that zGDNF<sup>mat</sup> binding is retained for all zGFRα1 constructs. Similar results were obtained in two other biological repeats. **b** Cut-away through the barrel crystal structure showing the location of K254E and L290E mutations within the zGFRα1<sup>D2-D3</sup> pentameric interface mapped onto a surface representation for one 2:2 complex. **c** Extracted raw particles of zGDNF<sup>mat</sup>-zGFRα1<sup>D2-D3+</sup> from negative stain electron microscopy (NS-EM) micrographs showing approximate five-fold symmetry. Representative images of a total of 336 raw particles showing this view. **d** Reference-free 2D class averages of zGDNF<sup>mat</sup>-zGFRα1<sup>D2-D3+</sup> from NS-EM images. Two dominant views are evident: a side view with clear density for each GDNF dimer stave and a top view showing the five-fold rotational symmetry of the GFRα1 pentameric subunits. The number of particles contributing to each 2D class average is listed below each class. Scale bar in (**d**, **e**): 10 nm. **e** Orthogonal views of NS-EM map from crosslinked zGDNF<sup>mat</sup>-zGFRα1<sup>D2-D3+</sup> sample either without (left) or with (right) the zGFRα1<sup>D2-D3+</sup>-zGDNF<sup>mat</sup> crystal structure fitted into the electron density using ChimeraX[77] "fit-in-map" tool.

experiments provide clear evidence that the GDNF-GFRα1 decamer can function as an adhesion complex on membranes.

**Formation of the *trans*-complex is disrupted by RET<sup>ECM</sup>**
We next asked whether the ability of GFRα1 to promote formation of the *trans*-adhesive complex is mutually exclusive with binding of RET, with which the GDNF-GFRα1 complex interacts *in cis* to promote trophic support[31,32]. Comparing the GDNF₂-GFRα1₂-RET₂ ternary structure with the GDNF₁₀-GFRα1₁₀ *trans*-adhesive barrel structure suggests that the GFRα1 binding sites for RET and homophilic GFRα1:GFRα1 interactions overlap, and revealed that GDNF is orthogonal to the cell membrane (Fig. 3d, Supplementary Fig. 4a, b). These observations suggest that the two structures are indeed mutually

exclusive. The high-affinity co-receptor binding site in the RET ternary complex involves GFRα1 D3 domain α15, its preceding loop and α12, which together form a wedge-shaped element to access the calcium binding site of RET[32] (Fig. 3d, Supplementary Fig. 4b). In the *trans*-synaptic GDNF-GFRα1 complex, the same wedge-shaped surface of GFRα1 D3 domain is central to the *cis* pentameric interface (Figs. 1d, 3d, Supplementary Fig. 4b).

To test the prediction that the two interaction modes of GFRα1 are mutually exclusive, we asked whether adding a soluble form of zRET<sup>ECM</sup> affects GDNF-dependent GFRα1 liposome adhesion. We preincubated zGFRα1<sup>D2-D3+</sup>-coated liposomes with soluble zRET<sup>ECM</sup>, and found that this completely abrogates the adhesive capacity of zGFRα1<sup>D2-D3+</sup> when exposed to zGDNF<sup>mat</sup> (Fig. 3a, green curve). We

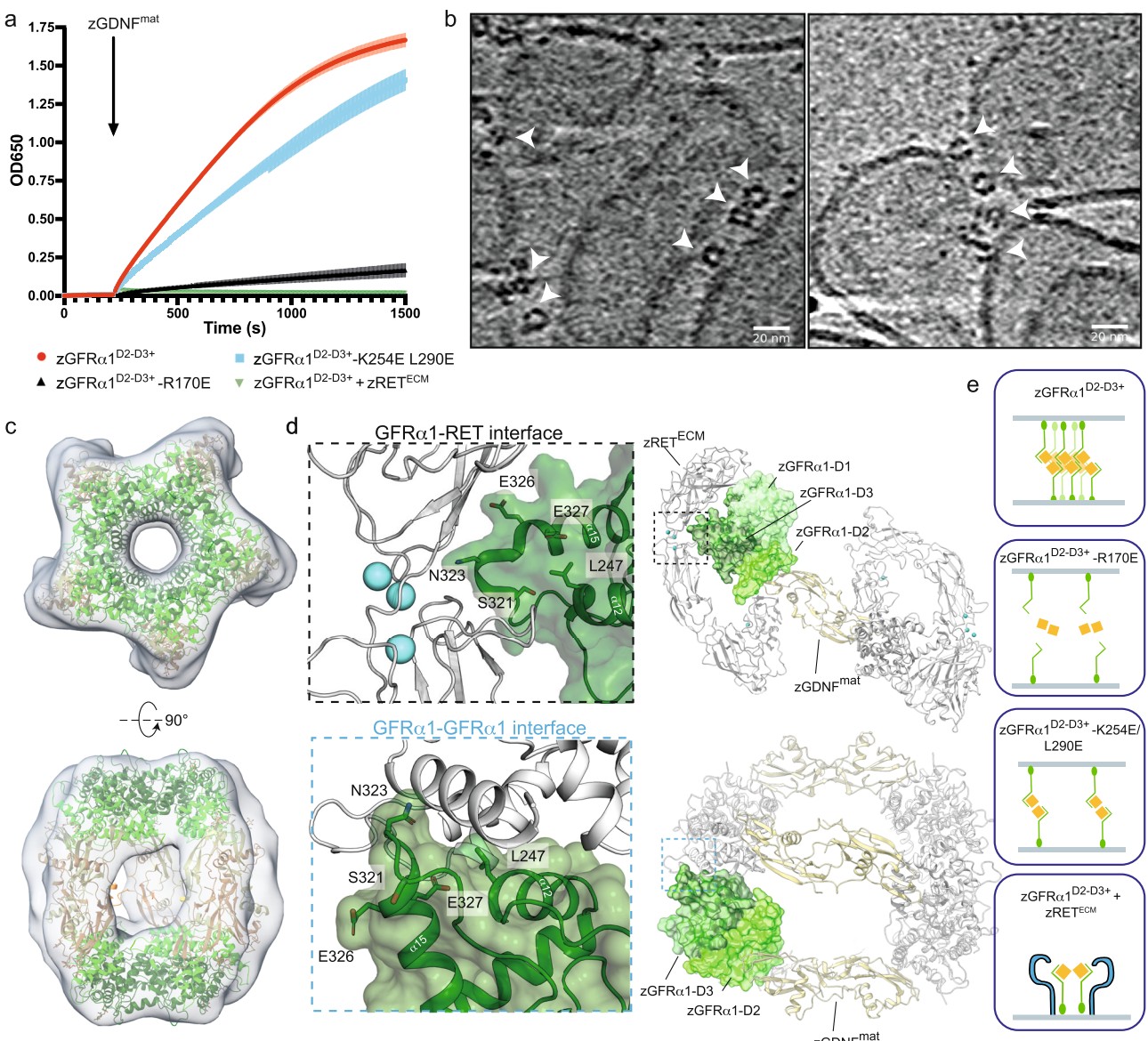

**Fig. 3 | Reconstitution of zGDNF-dependent *trans*-adhesion on liposomes is sensitive to pentameric interface mutation and addition of zRET extracellular module. a** Liposome adhesion assay using extruded liposomes coated with His-tagged zGFRα1 constructs as indicated. Liposome clustering was monitored by taking time-course measurements of absorbance at 650 nm (OD650). Arrow shows timepoint at which soluble zGDNF^mat was added. Error bars represent standard deviation from three technical repeats. **b** 2D tomographic slices from reconstructed tomograms of zGDNF^mat-zGFRα1^D2-D3+-mediated liposome aggregates. Images show close-up views of bridging protein density between two liposome membranes (indicated with white arrow heads). Scale bar: 20 nm. **c** Subtomogram map of the zGDNF^mat-zGFRα1^D2-D3+ adhesion complex bridging adhered liposomes with the decameric complex crystal structure fitted into the map using ChimeraX[77] 'fit-in-map' tool. Orthogonal views are shown. **d** Structural overlap between the high affinity GFRα1–RET interface and the GFRα1 pentameric interface. Top right, view of zGDNF-zGFRα1-zRET^ECM complex (PDB: 7AML) projecting down the two-fold molecular dyad. One zGFRα1 protomer is represented as a surface rendering

coloured according to domain, zGFRα1-D1 light green, zGFRα1-D2 green and zGFRα1-D3 dark green. Top left, close up view of zRET^ECM and zGFRα1 interaction at a calcium-junction-D3 domain interface. Selected interacting residues shown as sticks and calcium atoms shown as blue spheres. Bottom right, view of the pentameric interface from the decameric zGDNF^mat-zGFRα1^D2-D3+ structure highlighting a single zGFRα1 protomer by surface rendering. Bottom left, close-up view of the pentameric interface highlighting the same interacting residues as shown for RET interaction. **e** Schematic representation of the LiCAM capacity of GFRα1 under defined conditions. (i) zGFRα1^D2-D3+ acts as a strong adhesion molecule upon zGDNF^mat addition through the ability to form homophilic interactions *in cis*. (ii) R170E mutation in zGFRα1^D2-D3+ targets the ability of zGFRα1 to form interactions in trans and thereby abolishes the adhesive capacity of zGFRα1. (iii) Mutations targeting the *cis* pentamer interface, K254E L290E, lead to reduced adhesion function by zGFRα1 in the presence of zGDNF^mat. (iv) In the presence of zRET^ECM, zGFRα1^D2-D3+ preferentially forms a ternary complex *in cis* upon zGDNF^mat addition.

next established a liposome pelleting assay to determine how zRET^ECM and zGDNF^mat partition into membrane-associated and soluble fractions in order to assay their specific interactions with membrane-conjugated zGFRα1^D2-D3+. Soluble zGDNF^mat was found almost exclusively in the membrane-bound (pellet) fraction when incubated with zGFRα1^D2-D3+ (Supplementary Fig. 4c). This behaviour

was maintained for zGFRα1^D2-D3+ K254E L290E-coated liposomes, for which *cis* GFRα1 interactions are lost but GDNF binding is maintained. However, zGDNF^mat remained in the soluble fraction when the liposomes were coated with zGFRα1^D2-D3+ R170E, which has impaired GDNF binding (Supplementary Fig. 4c). Thus, an R170E mutation at the GDNF-GFRα1 interface abrogates GDNF binding to GFRα1 to

uncouple its LiCAM function. Further, soluble zRET$^{ECM}$ redistributes from the soluble to membrane-bound fractions in the presence of zGFRα1$^{D2-D3+}$ and zGDNF$^{mat}$, confirming the formation of a zGDNF$^{mat}$-zGFRα1$^{D2-D3+}$-zRET$^{ECM}$ ternary complex (Supplementary Fig. 4c). Together, these data reveal that zGFRα1 attached to liposomes spontaneously forms *trans*-adhesion complexes upon GDNF addition. However, the adhesive properties of zGDNF-zGFRα1 are blocked in the presence of soluble zRET$^{ECM}$, which redirects the system towards formation of *cis* zGDNF-zGFRα1-zRET ternary complexes (Fig. 3e).

We also evaluated the effect of soluble zRET$^{ECM}$ on formation of the *trans*-synaptic zGDNF$^{mat}$-zGFRα1$^{D2-D3+}$ complex in solution by monitoring the adhesive complex using native-PAGE. Here, adding soluble zRET$^{ECM}$ disrupted the ~700 kDa native PAGE band in a Ca$^{2+}$-dependent manner (Supplementary Fig. 4d). Calcium is known to be required for proper folding and transport of RET to the cell surface and for RET activation by GDNF[44,45]. We conclude that the dominant interaction between the RET calcium-binding site and GFRα1 D3 domain can disrupt pre-formed *trans*-adhesion assemblies by competing with the weaker pentameric *cis* interaction. Further, complete elimination of the ~700 kDa band is observed in the presence of HS (Supplementary Fig. 4d), indicating that RET and HS may synergise in disassembly of GDNF-GFRα1 *trans*-adhesive complexes. The impact of HS on the adhesive capacity of GFRα1 is discussed later.

## Cellular evidence for a *trans*-synaptic complex mediated by GFRα1 interaction

We next sought evidence from cellular assays as to whether the decameric GDNF-GFRα1 multimer seen bridging liposomes represents an adhesion complex that can bridge a synaptic gap of 20 nm[46]. We turned to a mammalian system for these studies, introducing mutations into rat GFRα1 based on our structural studies of the homologous proteins in the zebrafish GDNF-GFRα1 decameric complex. We first modified a previously-reported cell adhesion assay[4] to assess whether full length, GPI-anchored rat GFRα1 forms the same pentameric assemblies seen with the zebrafish proteins. GDNF was previously shown to induce clustering of GFRα1-expressing Jurkat cells, leading to classification of GFRα1 as a LiCAM[4]. We used HEK293 cells co-transfected with GFP and rat GFRα1 cDNA, which we refer to as mammalian GFRα1 full-length, or mGFRα1$^{FL}$ (Supplementary Table 2). The transfected HEK293 cells show a robust five-fold increase in adhesion upon rat GDNF stimulation as measured by the proportion of cell clusters seen after GDNF was added to mGFRα1$^{FL}$ expressing cells (Fig. 4a, b Supplementary Fig. 5a). As a positive control, cells were transfected with an established cell adhesion molecule, NCAM, which resulted in a comparable level of cell clustering that was independent of GDNF addition (Supplementary Fig. 5a).

Having established the basis for this assay, we next transfected HEK293 cells with the GFRα1 *cis* pentameric interface mutants. All of the mGFRα1 *cis* mutants retained some ability to confer GDNF-dependent increases in cell adhesion, and there was no significant difference in basal levels of adhesion between mutants and wild-type mGFRα1$^{FL}$ (Fig. 4b). However, each *cis* mutant supported significantly reduced levels of GDNF-induced cell clustering (Fig. 4b), and this did not reflect differences in expression levels as assessed by immunoblot analysis (Supplementary Fig. 5b). Moreover, immunofluorescence images indicated that the mGFRα1 *cis* mutants are all correctly processed and localised at the plasma membrane (Supplementary Fig. 5b). In addition, we used surface plasmon resonance (SPR) to quantify human GDNF$^{mat}$ binding by wild-type human GFRα1$^{25-424}$, GFRα1$^{150-424}$ (hereafter defined as hGFRα1$^{D1-CT}$ and hGFRα1$^{D2-CT}$ respectively) and the hGFRα1$^{D1-CT}$ *cis* mutants. No significant differences in steady-state affinity measurements were seen, (Supplementary Fig. 5c), confirming that the hGFRα1 *cis* mutants with an intact C-tail are correctly folded and functional.

Taken together, these data demonstrate that the GFRα1 pentameric *cis* interface seen in our zGDNF-zGFRα1 barrel structure is functionally relevant in mammalian GDNF-dependent cell adhesion. This finding further suggests that mammalian GFRα1 can form similar complexes. We note that the *cis* interface mutants show a weakened, but not a complete loss of adhesion function in the presence of GDNF (Fig. 4b). This was anticipated since the *trans*-adhesion interaction through a 2:2 GDNF-GFRα1 complex still remained intact. Thus, in the context of mGFRα1$^{FL}$, the capacity of GFRα1 to form pentameric interactions *in cis* strengthens the adhesion mediated by GFRα1 upon GDNF engagement in trans (Fig. 4c).

As previously published[4], deleting the N-terminal D1 domain of mGFRα1 has no impact on GDNF-dependent cell adhesion (Fig. 4b). This demonstrates that the mGFRα1$^{ΔD1}$ construct retains the elements required for GDNF-dependent cell adhesion and is functionally equivalent to mGFRα1$^{FL}$. This is consistent with the zGDNF-zGFRα1 barrel crystal structure and confirms our in vitro findings that GDNF-dependent adhesion complexes form in the absence of the D1 domain. Evidence that the presence of the D1 domain does not preclude adhesion complex formation in cellulo, unlike the in vitro data, also suggests the D1 domain may be sequestered at the cellular membrane.

To confirm the competition observed in vitro between formation of the GDNF$_{10}$-GFRα1$_{10}$ complex and the RET ternary complex in cells, we repeated the addition of human RET$^{ECM}$ to HEK293 cells producing mGFRα1$^{FL}$. We observed a significantly reduced proportion of cell aggregates when mGFRα1$^{FL}$-expressing cells were pre-incubated with RET$^{ECM}$ in the presence of GDNF (Fig. 4d). These data are consistent with the in vitro reconstitution experiments indicating that RET binding negatively regulates the GDNF-GFRα1 adhesive function.

## GDNF-driven adhesion in dendritic spine formation assay

To test the impact of *cis* interface mutations in a neuronal context, we next adapted a published dendritic spine formation assay for dissociated rat hippocampal neurons transfected with mGFRα1$^{FL}$ [42]. In this assay the functional consequence of GDNF-driven adhesion is to increase the number of dendritic spines (and therefore synaptogenesis). Briefly, dissociated rat hippocampal neurons were co-transfected with mGFRα1 interface mutants and GFP at 15 days after plating the cells (DIV15). The cultures were then treated with GDNF, prior to quantitative analysis of the density of dendritic spines (Fig. 4e, f, Supplementary Fig. 5d, e). This allowed us to evaluate the consequence of mGFRα1$^{FL}$ mutants in a synaptic compartment (postsynapse, dendrite) on the effect of endogenous mGFRα1 on the opposing compartment (presynapse, axon) to promote synapse formation. GDNF increases the density of dendritic spines in hippocampal neurons expressing mGFRα1$^{FL}$ (Fig. 4e, f), but this effect is reduced by *cis* mutations that target the GFRα1 pentamer interface (K251E, L287E and K251E-L287E). A mGFRα1 *trans* mutant (mGFRα1$^{Δl61}$), targeting the GDNF-binding interface[4,47], also showed a loss of spine density compared to wild-type mGFRα1$^{FL}$ (Fig. 4e, f). In contrast, mGFRα1$^{ΔD1}$-overexpressing neurons showed wild-type levels of GDNF-enhanced spine density (Fig. 4e, f). These data indicate that alongside the previously noted GDNF-dependent interactions in trans, the GFRα1 pentameric contribution *in cis* also promotes GDNF-driven spine formation in hippocampal neurons. Taken together, data from cell clustering and spine density assays argue that GFRα1 acts as a *trans*-synaptic organizing molecule in a manner that depends on the GFRα1 pentamer interface present in the GDNF$_{10}$-GFRα1$_{10}$ complex and required for the proper development of hippocampal connectivity.

## HS prevents *trans*-synaptic assemblies in a GFRα1$^{D1}$-dependent manner

Heparan sulphate (HS) has been shown to bind to the GFRα1 and GFRα2 receptors[19,22], raising the important question of whether HS also influences assembly of GDNF-GFRα1 *trans*-adhesion complexes. Cryo-

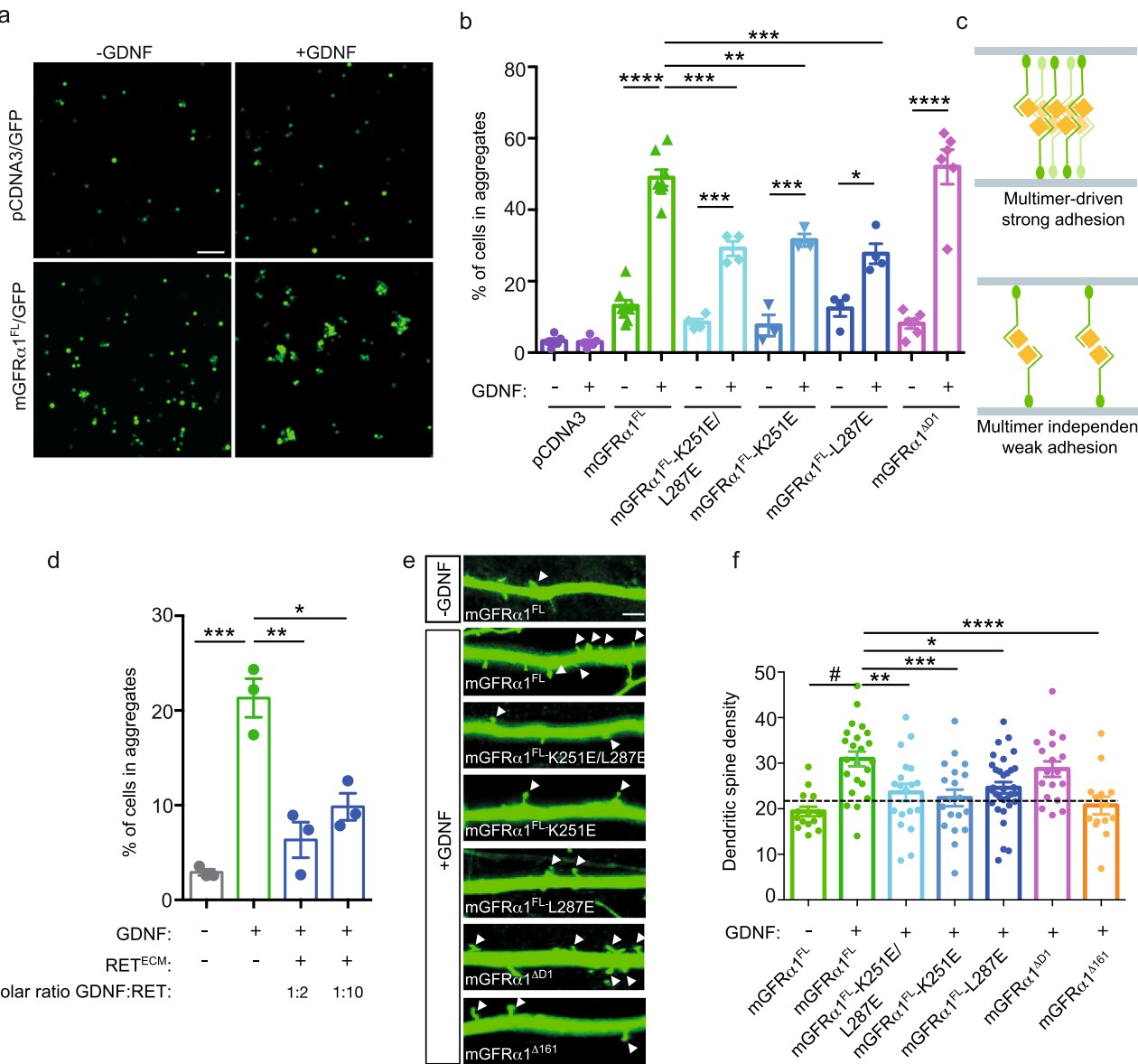

**Fig. 4 | Mammalian GFRα1 (mGFRα1) pentameric interface is required for GDNF-dependent adhesion in cell-clustering and neuronal dendritic spine assays. a** Cell adhesion assay using HEK293T cells transiently transfected with control pcDNA plasmid or mGFRα1^FL in the presence and absence of GDNF. White scale bar is 100 μm. **b** Effect of mutational analysis of mGFRα1 pentameric interface in HEK293T cell adhesion assay. The level of cell adhesion promoted by transfected proteins with and without GDNF treatment evaluated as the percentage of GFP+ cells present in aggregates more than 5 cells/field ± s.e.m. n ≥3 biologically independent experiments. **c** Schematic illustration to demonstrate impact of *cis* GFRα1 pentamer interface mutations on the ability of GFRα1 to act as a cell adhesion molecule. GPI-anchored GFRα1 coloured green and soluble GDNF yellow. For GFRα1 wild-type, adhesion is driven through both GDNF-dependent interactions in trans and GFRα1 homophilic contribution *in cis* (top). For GFRα1 mutants that target the pentamer interface, GDNF mediates adhesion through bridging GFRα1 molecules in trans (bottom). **d** The presence of soluble RET^ECM interferes with GDNF-induced adhesion of GFRα1-expressing HEK293T cells. HEK293T cells

expressing mGFRα1 and GFP were preincubated with human RET^ECM for 2 h in the absence of GDNF. GDNF was then added for an additional 2 h at room temperature. The bar graph indicate the percentage of GFP⁺ cells in aggregates greater than 5 cells ± s.e.m. *n* = 3 biologically independent experiments. **e** In the presence of GDNF, mGFRα1 mutants reduce spine density of dissociated hippocampal neurons. Hippocampal neurons transfected with GFP-expressing plasmid in combination with indicated constructs at 15 DIV, maintained in the absence or presence of GDNF as indicated in the figure for 72 h. Arrows indicated dendritic spines along the dendritic shaft. Scale bar: 50 μm. **f** Quantification of total dendritic spines along 100 μm of dendritic length of hippocampal neurons. Mean ± s.e.m. from three independent experiments (*n* = 14–32 neurons/condition). Dashed line indicates dendritic spine density on neurons transfected with empty vector cultured in the absence of GDNF. **b** *\*p* = 0.0154, \*\**p* = 0.0016, \*\*\**p* = 0.0002, \*\*\*\**p* < 0.0001; (**d**) \**p* = 0.0037, \*\**p* = 0.007, \*\*\**p* = 0.0002; (**f**) \**p* = 0.03, \*\**p* = 0.0236, \*\*\**p* = 0.0041, \*\*\*\**p* = 0.0012, ^#*p* < 0.0001. One-way ANOVA, followed by Tukey´s multiple comparison test.

EM and crystal structures of GFRα1 and GFRα2 suggest that the tight binding site for HS is formed by the D1 domain together with D2-D3 module, contributing an extended basic surface to engage the negatively charged HS[19,22,32] (Fig. 5a). We first asked whether the GFRα1 D1 domain contributes to the GFRα1 HS binding site by measuring HS binding affinities for different truncations of human GFRα1 (hGFRα1)

using isothermal calorimetry (ITC). Representative plots are shown for hGFRα1^25–424 (hGFRα1^D1-CT) and hGFRα1^150–424 (hGFRα1^D2-CT, lacking the amino-terminal D1 domain) using HS with a degree of polymerisation of 10 (dp10) (Fig. 5c, Supplementary Table 3) with equilibrium binding constants determined as 82 ± 49 nM (D1-CT) and 12.4 ± 1.9 μM (D2-CT) respectively. The 150-fold weaker binding affinity seen for hGFRα1^D2-CT

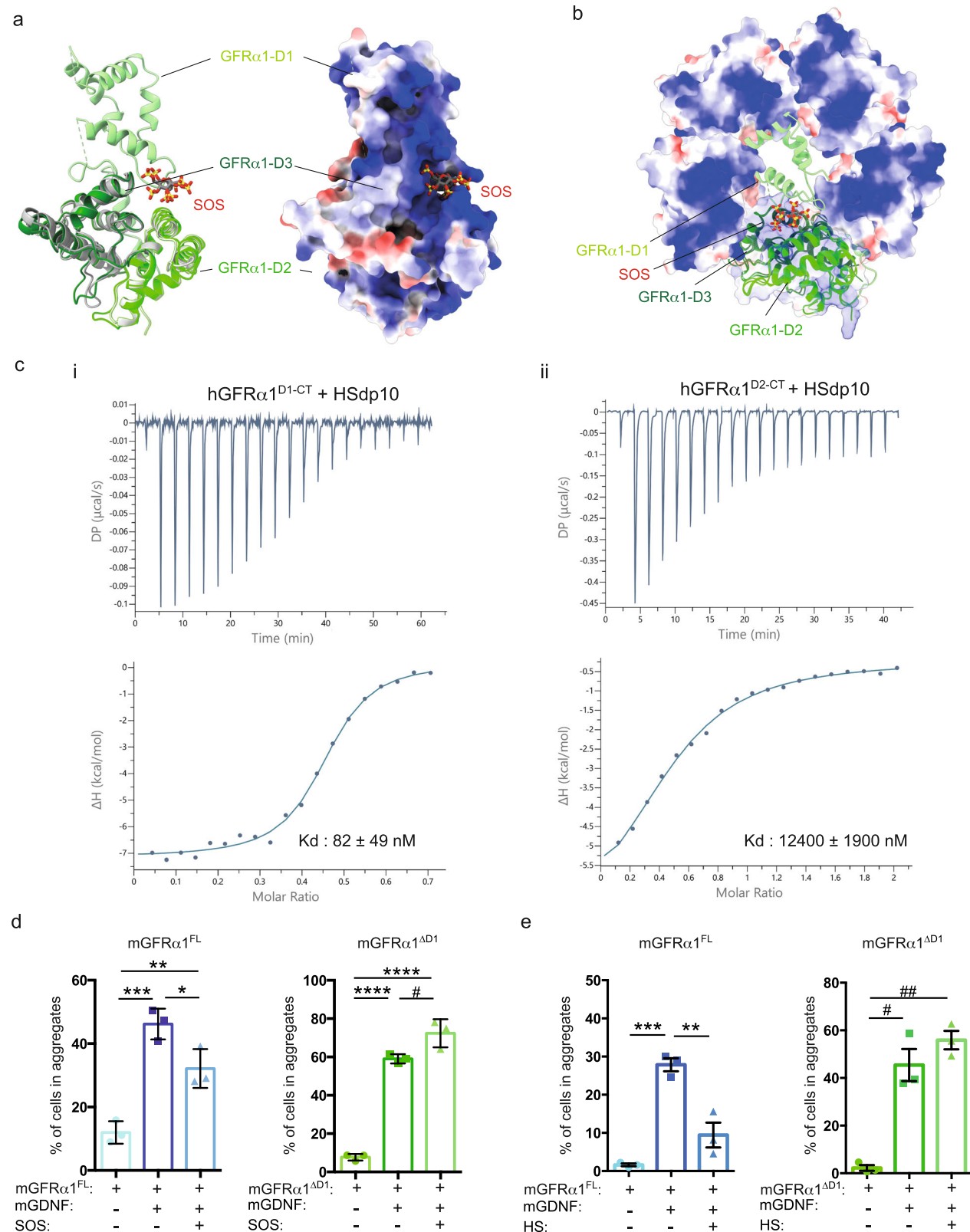

provides clear biophysical evidence that the GFRα1 D1 domain contributes substantially to the HS binding site. A similar trend in the binding affinities between hGFRα1$^{D1-CT}$ and hGFRα1$^{D2-CT}$ with a HS chemical mimetic, sucrose octasulfate (SOS), was also observed (Supplementary Fig. 6a, Supplementary Table 3). Further, the derived stoichiometry of the interaction between hGFRα1$^{D1-CT}$ and HS was 0.41 (Supplementary Table 3) implying that a single HS dp10 chain can

bridge two GFRα1 molecules. The derived stoichiometry for hGFRα1$^{D2-CT}$ and HS of 0.50 indicates a similar binding mode in the absence of the D1 domain and therefore rules out the possibility that the D1 domain contributes to binding through a second discrete HS binding site.

To identify whether GFRα1 can simultaneously engage both HS and form a *trans*-adhesive assembly, zGFRα1$^{D1-D3}$ (PDB: 7AML), mGFRα1$^{D2-D3}$-SOS (PDB: 2V5E) and zGFRα1$^{D2-D3}$ pentamers were

**Fig. 5 | D1 domain-dependent binding of sulfated glycosaminoglycans (GAGs) disrupts *trans*-synaptic GDNF-GFRα1 complexes. a** Structural superimposition of the mGFRα1$^{D2-D3}$-SOS complex (PDB: 2V5E) with an intact zGFRα1$^{D1-D3}$ (PDB: 7AML) shown as a ribbon representation (left) and a surface rendering coloured according to Columbic electrostatic potential (right). mGFRα1$^{D2-D3}$ cartoon coloured grey with D2 light grey and D3 in dark grey. zGFRα1$^{D1-D3}$ cartoon coloured in green with D1 light green, D2 green and D3 dark green. SOS ligand, shown as sticks, binds to a positively charged cleft formed between all three domains. **b** Steric clash between D1 domain of zGFRα1$^{D1-D3}$ (PDB: 7AML) and zGFRα1$^{D2-D3}$ pentamers within the decameric complex. zGFRα1$^{D2-D3}$ pentamer shown as a surface representation coloured according to electrostatic surface potential. mGFRα1$^{D2-D3}$-SOS complex (PDB: 2V5E) is superposed onto zGFRα1$^{D1-D3}$ shown as cartoon coloured in green with D1 light green, D2 green and D3 dark green. Superimpositions were done in UCSF ChimeraX using the MatchMaker tool[85]. **c** ITC analysis of hGFRα1 interactions with HS. (i) hGFRα1$^{D1-CT}$ binding to HS dp10 and (ii) hGFRα1$^{D2-CT}$ binding to HS dp10.

Raw ITC titration data plotted against time (top) and integrated heat signals plotted as a function of molar ratio (bottom). Circles represent the integrated heat of interaction, while blue curves represent the best fit obtained by non-linear least-squares procedures using the "One set of sites" model. Representative titrations and binding curves are shown. Derived binding constants (Kds) are reported on each plot, as mean values of ≥3 independent experiments ± standard deviation. **d**, **e** D1 domain impact of GAGs binding to mGFRα1 in the HEK293 adhesion assay. HEK293T cells were transfected with vector alone, mGFRα1$^{FL}$ or mGFRα1$^{ΔD1}$ with GFP. GFP-expressing cells were preincubated with SOS (**d**) or HS (**e**) for 2 h in the absence of GDNF. GDNF was then added for an additional 2 h at room temperature. The percentage of cells in aggregates greater than 5 cells under the indicated conditions is shown. Mean values of triplicate experiments ± s.e.m. (**d**) \*$p = 0.0306$, \*\*$p = 0.0059$, \*\*\*$p = 0.0004$, \*\*\*\*$p < 0.0001$, #$p = 0.0268$, (**e**) \*$p = 0.021$, \*\*\*$p = 0.0003$, #$p = 0.0013$, ##$p = 0.004$. One-way ANOVA, followed by Tukey's multiple comparison test.

---

superimposed (Fig. 5b). The superpositions anticipate that full-length GFRα1 bound to a GAG, with the D1 domain secured above the D3 domain, would sterically occlude the formation of GFRα1 pentamers required for the *trans*-adhesive function of GDNF-GFRα1.

We tested this prediction by determining the impact of HS binding to GFRα1 on the GDNF-GFRα1 LiCAM function. To do this we used the HEK293 cell-based adhesion assay transfected with either mGFRα1$^{FL}$ containing a fully functional HS/SOS site or mGFRα1$^{ΔD1}$ that has significantly reduced HS binding capacity. Both populations of transfected cells were pre-treated with exogenous HS or SOS for 2 h at room temperature prior to GDNF addition. The percentage of cells that formed clusters was then quantified as described earlier. Consistent with our structural superpositions, for mGFRα1$^{FL}$-expressing cells pretreatment with SOS or HS led to a significant reduction in the number of GDNF-induced cell clusters compared to the control (Fig. 5d, e, Supplementary Fig. 6b). By contrast, neither SOS nor HS reduced the cell clustering capacity for mGFRα1$^{ΔD1}$-transfected cells (Fig. 5d, e, Supplementary Fig. 6b). These data indicate that binding of SOS or HS to GFRα1 can inhibit the LiCAM function of GDNF-GFRα1 in a D1-dependent manner. We propose that this modulation arises through HS/SOS blocking of GFRα1 *cis* contributions (in the 10:10 *trans*-adhesive complex) by coupling of the GFRα1 D1:D3 interface. Interestingly HS/SOS treatment had no effect on pre-formed GDNF-driven mGFRα1-mediated cell clusters (Supplementary Fig. 6c), indicating that upon formation of an adhesion complex HS proteoglycans are unable to dismantle pre-assembled *trans*-adhesive GDNF-GFRα1 complexes under these conditions. This demonstrates that GFRα1 LiCAM capability is only sensitive to regulation by HS proteoglycans if they are present prior to the secretion of and availability of GDNF.

## Discussion

The mechanistic basis for GDNF-dependent cell adhesion is unexplained despite the importance of understanding GDNF function as a promising treatment for Parkinson's disease. Here we identify a multivalent assembly comprised of GDNF-GFRα1 subunits and describe its architecture. We reconstitute the assembly on liposomes, demonstrating its adhesion properties and show how two regulatory partners RET and HS prevent complex assembly. Our results challenge the prevailing view that the GFRα1 receptor for GDNF family ligands acts simply as a passive co-receptor for RET signalling. Instead our data implicate GFRα1 as an active signalling integrator that directs control of GDNF-dependent *trans*-adhesion, synapse maturation as well as trophic support (Fig. 6).

Identification of the decameric GDNF-GFRα1 adhesive assembly has several implications for understanding GDNF function. This complex presents GDNF dimers perpendicular to the cell membrane ("staves" of the barrel) in contrast to a parallel arrangement observed for GDNF-GFRα1-RET$^{ECM}$ trophic complexes[31,32,48]. The key to these different arrangements is likely the highly flexible GFRα1 C-tail that

allows different co-receptor conformers to present GDNF homodimers in two opposing arrangements. We hypothesize that RET interaction biases GDNF signalling towards *cis* trophic support by forming a pre-assembled complex with GFRα1[49,50], orienting the GDNF-binding site parallel to the cell membrane, and blocking the GFRα1:GFRα1 pentameric interace. In the absence of RET, GFRα1 instead adopts a conformation with its GDNF-binding site projecting towards a second cell membrane, thus promoting GDNF-dependent adhesion and the assembly of *trans* GDNF-GFRα1 multimers into the barrel shown in Fig. 1. The different roles for GFRα1 depending on the presence of RET may reflect a mechanism for switching from synaptic adhesion to trophic support during maturation of newly-formed synapses in neurons that express GFRα1 and RET. These include dopaminergic neurons[27,51], motor neurons[52–54] and neurons of the peripheral system[55,56]. RET cell surface expression is known to be controlled by calcium influx, which could enhance RET folding and transport[45]. However, the synaptic adhesion function of GFRα1 has been reported in hippocampal and cortical neurons that express GFRα1 but lack RET[4,42,57]. Thus, an alternative explanation is that adhesion and trophic support reflect two different GDNF functional outcomes depending on whether RET is present or absent at the neuronal synapse.

We also show that HS influences assembly of the decameric complex in vitro and in cells. HS binding at the D1-D2-D3 junction has a disruptive impact on the GDNF-GFRα1 adhesion complex in a D1-dependent manner. HS binding likely couples the D1:D3 interface, promoting a GFRα1 conformation that sterically occludes the assembly of the pentameric subunits of the adhesion complex (as shown by the structural superimposition of zGFRα1$^{D1-D3}$ with the zGFRα1$^{D2D3}$ pentamer subunit). Competition between direct HS binding and the formation of *trans*-synaptic complexes has also been reported previously for LAR-RPTP/IL1RAPL1[58] and RPTPσ/TrkC[59] adhesion systems. In these examples, the dominant binding of HS disrupts pre-assembled adhesion complexes in solution and in cell-based assays[58,59]. Binding of HS to synaptic receptors may thus be a common regulatory mechanism to prevent the assembly of adhesion complexes.

Proteoglycans are highly enriched at the synaptic cleft, forming a synaptic proteoglycan layer[60–62] and an RNA sequencing study has demonstrated cell-type specific expression patterns for HS proteoglycans (HSPGs) in primary hippocampal neurons[63]. Heterogeneity in HSPG expression may also exist at the level of specific dendrites and axons within the same neuron[64]. This indicates that there might be distinct neuronal contexts in which GFRα1 can escape the proteoglycan layer and promote adhesion. Alternatively, we cannot exclude a proteolytic mechanism involving an ADAM-dependent clipping of the GFRα1-D1 domain that could desensitise GFRα1 to HSPGs at the synapse and thereby promote the LiCAM function.

A common characteristic of synaptic adhesion mechanisms involving membrane-bound soluble factors is the cooperative interplay between *cis* interactions and adhesive *trans* contacts[65]. The

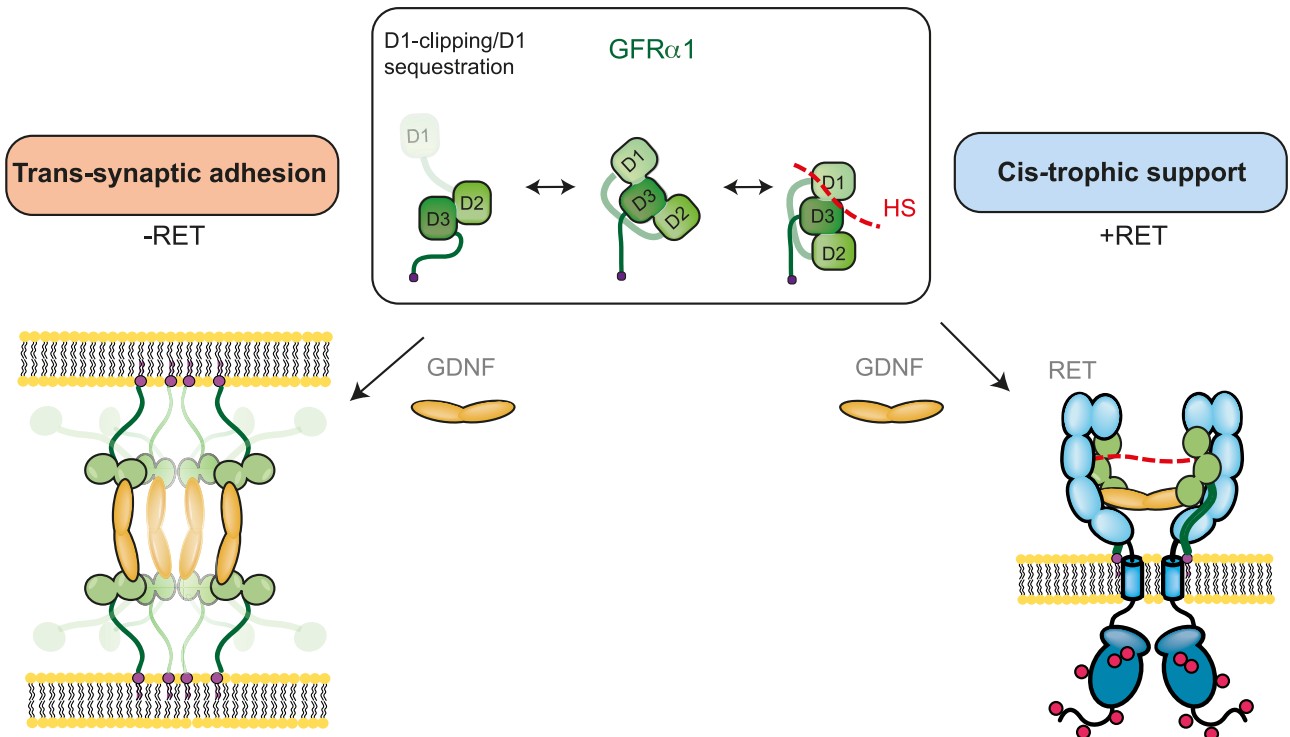

**Fig. 6 | A model for GFRα1-directed conformational control of GDNF-dependent signalling.** GFRα1 mediates *trans*-synaptic adhesion following GDNF engagement. D1 proteolytic clipping or D1 sequestration by a GFRα1 binding partner at the cell membrane promotes the formation of a GDNF-GFRα1 decameric *trans*-synaptic complex. However, in the presence of RET and HS, GFRα1 forms a ternary complex with RET *in cis* following GDNF engagement, to promote trophic support. RET and HS binding couples the D1:D3 interface and thereby prevents *trans*-adhesion complex formation.

decameric assembly in solution follows the formation of 2:2 GDNF-GFRα1 complexes that individually potentially mimic *trans* interactions between the pre- and postsynaptic membrane. Multimerisation in solution or through anchoring to a physiologically relevant membrane is likely driven through the weaker GFRα1 interactions *in cis* to assemble the GDNF-GFRα1 decamer. We cannot distinguish whether assembly precedes via preformed GFRα1 pentamers on each adhering liposome membrane subsequently being coupled by dimeric GDNF ligand or through the multimerization of *trans* 2:2 GDNF-GFRα1 complexes. Further work aiming to capture assembly intermediates towards the decameric state may inform on the precise assembly pathway used.

In this study we show that the presence of the D1 domain impedes decameric complex formation in vitro but does not prevent formation in cellulo. This is consistent with packing of the D1 against D3 sterically perturbing the formation of the pentameric rings based on structural comparisons. Relieving D1 domain-mediated antagonism of adhesion could arise either by proteolytic clipping (as observed in vitro) or by D1 displacement from domain D3 contact (D1 sequestration) promoted by GFRα1 binding partners in cellulo. The ability of GFRα1 to adopt an alternative conformation at the cell-membrane would be consistent with our cell-based data. Indeed evidence for a D1-specific GFRα1-binding partner such as NCAM has been published[66]. NCAM has been demonstrated to mediate GDNF-GFRα1 induced post synaptic maturation in hippocampal neurons[42] and is present on both pre- and postsynaptic membranes[42,67,68]. We speculate that NCAM or other unknown partners for the D1 domain may positively impact GDNF-driven adhesion in an opposite manner to RET and HS. However, whether HS peturbs NCAM interaction with the D1 domain is not known and is outside the scope of this study.

In summary, we identify a multivalent assembly of GDNF-GFRα1 with adhesion and synaptogenic properties. We reconstituted the assembly from purified components confirming a liposome adhesion function and validated assembly interfaces using two independent cell-based assays. Finally, we uncovered two regulatory partners RET and HS that suggest a mutually exclusive relationship between GDNF-directed adhesion and trophic support that is dependent on their precise cellular context.

## Methods

### Protein expression and purification

Expression constructs encoding zGFRα1$^{20–368}$ (zGFRα1$^{D1-D3+}$) and zGFRα1$^{144–368}$ (zGFRα1$^{D2-D3+}$) were PCR amplified from a zGFRα1a cDNA gift from Dr Ian Shepherd (Emory University, NCBI accession code AAK11260). zGFRα1$^{D2-D3+}$-K254E L290E and zGFRα1$^{D2-D3+}$-R170E variants were produced by overlap extension PCR mutagenesis. hGFRα1$^{25–424}$ (hGFRα1$^{D1-CT}$) and hGFRα1$^{150–424}$ (hGFRα1$^{D2-CT}$) were subcloned into a modified pCEP-Pu vector containing an N-terminus BM40 secretion sequence and a C-terminal His$_6$-tag[69]. The coding sequences for the mature structured region of zGDNF (residues 135-235: zGDNF$^{mat}$) and the structured region of hGDNF (residues 110-211: hGDNF$^{mat}$) were also ligated into a pCEP-Pu vector containing an N-terminus BM40 secretion sequence followed by a N-terminal His$_6$-tag and TEV protease cleavage site. Each of these proteins was expressed as a secreted protein in Expi293F cells (ThermoFisher, Cat #A14527), except for the zGDNF$^{mat}$–zGFRα1$^{D1-D3}$ complex, which was expressed and purified as previously described[32] using Sf21 insect cells as an expression host. Purification of soluble human and zebrafish RET extracellular modules were performed as described in Adams et al.[32]. A summary of the proteins used for each set of experiments is shown in Supplementary Table 1. For Expi293F cell expression, cells were grown in serum-free Freestyle 293 Expression Medium to a cell density of $1.5 \times 10^6$/ml prior to transfection with the vectors using linear polyethylenimine (Polysciences). Transfected cells were then incubated for 96 h at 37 °C, 8% $CO_2$ with continuous shaking at 125 rpm before cell culture supernatant was harvested by centrifugation at 2500 x $g$ at

4 °C. The supernatants were pooled and adjusted to a final concentration of 20 mM Tris (pH 8.0) and 10 mM imidazole. Ni$^{2+}$-NTA agarose (Qiagen) was added in batch to the conditioned media for 2 h at 4 °C with continuous rotation. The resin was recovered and the bound protein washed with 20 mM HEPES (pH 7.5), 500 mM NaCl and 10 mM imidazole. Bound protein was then eluted with three incubations of 20 mM HEPES (pH 7.5), 150 mM NaCl, 500 mM imidazole for 15 min at 4 °C with continuous rotation. For zGDNF$^{mat}$ and hGDNF$^{mat}$, resin-bound protein was eluted by overnight cleavage with TEV protease. Eluents were pooled and concentrated before further purification using size exclusion chromatography using a Superdex 200 Increase 10/300 GL column (GE Healthcare) in 20 mM HEPES (pH 7.5), 130 mM NaCl. Peak fractions were pooled and concentrated. For co-transfections to assemble GDNF-GFRα1 co-complexes fractions were pooled and concentrated to 3–4 mg/ml.

## Crystallisation and structure determination for zGDNF$^{mat}$–zGFRα1$^{D1-D3}$

Purified zGDNF$^{mat}$-zGFRα1$^{D1-D3}$ was concentrated in 20 mM Tris (pH 7.0), 100 mM NaCl and 1 mM CaCl$_2$ to 2.5 mg/ml. Vapour diffusion experiments were set up in sitting drop trays (MRC-2 drop trays) at 22 °C using a Mosquito robot (TTP LabTech). The reservoir precipitant solution was 100 mM Tris (pH 7.5), 3% (v/v) acetonitrile, 5% (w/v) PEG 20, 000 and 100 mM NaCl. 600 nl drops were prepared by mixing 300 nl protein, 200 nl reservoir precipitant solution and 100 nl of microseeds. Crystals appeared after 34 days and were flash-frozen in liquid nitrogen and cryoprotected in 30% (v/v) ethylene glycol for data collection. Crystals belong to the monoclinic P2$_1$ space group with cell constants a = 114.1 b = 170.0 c = 130.8 Å and β = 96.2°. A Matthews coefficient ($V_M$) of 3.46 Å$^3$/Da with 65% solvent content suggested ten copies of rGFRα1$^{D2-D3}$ and hGDNF$^{mat}$ (PDB: 3FUB) within the asymmetric unit.

X-ray data were collected at IO4-1 beamline at the Diamond Light Source using a DECTRIS PILATUS 6 M detector. Data integration and reduction was performed using DIALS (v.2.2.5), implemented within the xia2 pipeline programme[70]. Molecular replacement was performed by PHASER (v.2.8.3)[71] and identified 10 copies of zGDNF$^{mat}$-zGFRα1$^{D1-D3}$ using hGDNF$^{mat}$-rGFRα1$^{D2-D3}$ (PDB code: 3FUB) as a search model. Each solution was transformed back into the same asymmetric unit to reconstruct the decameric complex. The structure was built and refined in COOT v0.9.8.6[72,73] and PHENIX.REFINE (v.1.20.1_4487)[74,75]. Glycosylation sites were validated using PRIVATEER (MkIV)[76]. The final refined model had R-factor 23.8% and Rfree of 28.0%. A Ramachandran plot shows 96.48% in favoured region, 3.49% allowed and 0.03% outliers regions.

Structure-based images for the figures were rendered in UCSF ChimerX (v.1.4)[77] and Pymol (v.2.4)[78]. Structural superimpositions were performed in Pymol (v.2.4)[78].

## Native-PAGE analysis of zGDNF$^{mat}$-zGFRα1 complexes

Blue native polyacrylamide gel electrophoresis (BN-PAGE) was used to separate protein complexes without denaturation[79,80]. Following Ni$^{2+}$-NTA affinity purification, 10 µl of zGDNF$^{mat}$-zGFRα1 sample at 2 mg/ml in 10 mM HEPES (pH 7.5), 130 mM NaCl, 1 mM CaCl$_2$ was incubated with 4 x native page loading buffer (Invitrogen) and loaded into a 3-12% 10 well Bis-Tris NativePAGE gel (Invitrogen). zRET$^{ECM}$, HS dp10 (100 µM) (Iduron) and EDTA (2 mM) were pre-incubated with samples for 1 h at 4 °C prior to the addition of loading buffer. NativePAGE™ running buffer (Invitrogen) was added to the anode chamber and running buffer supplemented with NativePAGE™ Cathode Buffer Additive (Invitrogen) was added to the cathode chamber. Gels were run for 90 min at 150 V. NativeMark unstained protein molecular weight standards (Invitrogen) were run alongside protein samples. Gels were destained with 40% (v/v) methanol, 10% (v/v) acetic acid.

## zGDNF$^{mat}$-zGFRα1$^{D2-D3+}$ negative-stain EM data acquisition and processing

To prepare zGDNF$^{mat}$-zGFRα1$^{D2-D3+}$ complex for negative stain EM, the sample was cross-linked with 0.1% glutaraldehyde for 30 min at room temperature. The reaction was quenched by adding 1 M Tris (pH 7.5) to a final concentration of 75 mM Tris (pH 7.5). The sample was then diluted by adding 300 ml of 20 mM HEPES (pH 7.5), 130 mM NaCl before further purification by size-exclusion chromatography using a Superose 6 Increase 10/300 GL column (Sigma-Aldrich) in 20 mM HEPES (pH 7.5), 130 mM NaCl. Fractions of interest were analysed on a reducing SDS-PAGE gel. Negative stain EM grids were prepared by glow discharging carbon-coated 200-mesh copper grids (C200Cu100EM) at 45 mA for 45 s using a PELCO EasiGlow discharge unit. Fraction C1 (Supplementary Fig. 2c) from size-exclusion chromatography of the cross-linked sample was diluted 1:3 with 20 mM HEPES pH 7.5, 130 mM NaCl before 4 µl was applied to freshly glow-discharged grids for 60 s. Excess sample was removed by blotting with filter paper, before the grid was briefly washed by placing carbon-side down in 10 µl of 2% (w/v) uranyl acetate solution (Agar Scientific). Stain was then removed by further blotting and the grid placed again carbon-side down on a second 10 µl drop of 2% (w/v) uranyl acetate solution. Stain was removed by a final blot before the grid was left to dry.

Negative stain EM micrographs were collected on a Tecnai Twin T12 microscope (Thermo Fisher) operated at 120 kV and equipped with BMUltrascan 1000 2048×2048 CCD detector. A total of 540 micrographs were collected with a defocus of −1.5 µm and at nominal magnification of x 30,000 yielding a pixel size of 3.45 Å/px. Semi-automated particle picking from raw micrographs was performed using Xmipp (v.3.0)[81] and subsequently 52,913 particles were extracted in RELION (v.3.1)[82,83]. 2D classifications were performed in RELION (v.3.1)[82,83] to remove poor particles (noisy, featureless particles) and generate final well-resolved 2D class averages. 5,221 particles were used to generate an ab initio model in Relion (v.3.1)[82,83] that was low-pass filtered to 60 Å and used as a reference map for 3D classifications performed in RELION (v.3.1)[82,83]. 3,132 particles were selected for the final reconstruction using RELION (v.3.1) refinement protocol[82,83] resulting in a final reconstruction at a global resolution of 30 Å using the gold-standard FSC. All image processing was performed in Scipion (v.3.0)[84]. Fitting of the crystal structures into the electron density map was performed in UCSF ChimeraX (v.1.4)[77,85] using the 'fit-in-map' tool. Images of maps were produced in UCSF ChimeraX (v.1.4)[85] and structure-based images rendered in UCSF ChimeraX (v.1.4)[77].

## Liposome preparations and zGDNF$^{mat}$-zGFRα1$^{D2-D3+}$-mediated liposome adhesion assays

Prior to evaporation of solvent and drying of lipid film, lipid mixtures containing a 9:1 molar ratio 1,2-dioleoyl-sn-glycero-3-phosphocholine (DOPC): 1,2-dioleoyl-sn-glycero-3-[(N-(5-amino-1-carboxypentyl)iminodiacetic acid)succinyl] (DGS-NTA) (Avanti) were prepared. The solvent was removed using a continuous stream of argon followed by overnight drying in a rotary evaporator vacuum system. Dried lipid films were rehydrated in assay buffer, 20 mM HEPES pH 7.5, 180 mM NaCl, 1 mM CaCl$_2$ for 1 h at 37 °C, with vortexing every 5 min. Hydrated lipid mixtures were then passed through a Mini-Extruder (Avanti) with a polycarbonate membrane with a pore size of 100 nm. The size distribution of prepared liposomes was assessed by dynamic light scattering measurements on a Malvern Zetasizer Nanoseries.

For the adhesion assay, His-tagged zGFRα1$^{D2-D3+}$ was conjugated to liposome surfaces by incubating His-tagged zGFRα1$^{D2-D3+}$ (3 µM) with lipids (3 mM) in assay buffer at room temperature for 30 min. Samples were then loaded into a QS.1.0 cuvette and absorbance measurements at 650 nm (OD650) were taken on a UV-VIS 550 spectrometer (Jasco) at 18 °C. Absorbance measurements were taken with a time interval of 5 s and for a total of 25 min. zGDNF$^{mat}$ (3 µM) was added after 220 s. For adhesion experiments in the presence of human RET$^{ECM}$, zGFRα1$^{D2-D3+}$-

coated liposomes were preincubated with RET$^{ECM}$ (3 µM) for 30 min at room temperature. Samples were run in triplicate and the average OD650 measurement was plotted for each time-point with error bars showing ± standard deviation.

## Cryo-ET studies of zGDNF$^{mat}$-zGFRα1$^{D2-D3+}$ bridging complexes between liposome membranes

zGDNF$^{mat}$-zGFRα1$^{D2-D3+}$-mediated liposome aggregates were prepared as described for the liposome adhesion assay and incubated at room temperature for 1 h. 4 µl of liposome aggregates were mixed with a 1:3 volume ratio of protein A-coated 10 nm gold fiducial beads (BBI solutions) and applied to freshly glow discharged R 2/2 300 mesh Cu Quantifoil$^{TM}$ grids at 25 °C. The sample was then blotted for 3 s before being plunge-frozen in liquid ethane using a Vitrobot Mark IV (Thermo Fisher). A cryo-ET data collection of frozen-hydrated zGDNF$^{mat}$-zGFRα1$^{D2D3+}$ aggregated liposomes was carried out on a Talos FEI Artica operated at 200 kV and equipped with a Falcon 3 detector at the Francis Crick Institute. Data acquisition was conducted using Tomography software (v.5.12) (ThermoScientific). A single 2D tilt series was collected per hole, using a dose-symmetry angular acquisition scheme from −57° to 57° with a 3° tilt increment. 2D projections were captured at a nominal magnification of ×45,000 that resulted in a pixel size of 3.255 Å / px. A total of 12 tilt series were collected with a defocus range of −4 to −7 μM. A single frame was taken per tilt angle, with a total electron dose of 77.61 e⁻/Å² per tilt series, which was fractionated to 1.99 e⁻/Å² per tilt image, and with a dose rate of 29.25 e⁻/px/s. 3D tomogram reconstructions were generated using the IMOD suite of programmes (v. 4.12)[86]. The final alignments were down-sampled by a factor of 4 and final tomograms were generated using a back-projection algorithm with a SIRT-like filter equivalent to 5 iterations of the SIRT algorithm. 2D tomographic slices were generated in 3dmod within the IMOD suite (v.4.12)[86].

For subtomogram averaging, 502 particles between liposome membranes were picked manually from 9 reconstructed tomograms using 3dmod within the IMOD suite (v.4.12)[86]. Particles were picked from tomograms reconstructed at a binning factor of 4 and with a SIRT-like filter applied. Contrast transfer function (CTF) parameters of each the tilt-series was estimated using CTFFIND (v4.1)[87] and subtomogram averaging was performed using RELION (v.4.0)[82,88,89]. Raw tilt series and particle coordinates were imported into RELION (v.4.0)[82,88,89] and pseudo-subtomogram particles were generated with an un-binned 80 pixel box size. 3 ab initio models were generated from pseudo-subtomograms using the de novo 3D model generation programme (using a stochastic gradient descent algorithm) in RELION (v.4.0)[82,88,89]. The highest quality model (class III, Supplementary Fig. 4c), generated from 143 particles, had structural features closely resembling the decameric crystal structure (five-fold symmetry and bridging staves between pentameric rings). This model was low pass filtered to 40 Å and used as a reference model for 3D classification performed in RELION (v.4.0)[82,88,89] with 2 classes, without the imposition of symmetry, and using a spherical mask of 180 Å. Class I, containing 187 particles (Supplementary Fig. 4c), displayed the highest resolution features, and these particles were subjected to 3D autorefinement, with either C1 or D5 symmetry imposed. The final zGDNF$^{mat}$-zGFRα1$^{D2-D3+}$ adhesion complex map, reconstructed with D5 symmetry imposed, gave a resolution of 22 Å as calculated using a soft spherical mask and the gold-standard FSC. Fitting of the crystal structure into the subtomogram average was performed in UCSF ChimeraX (v.1.4)[77,85] using the 'fit-in-map' tool. Structure-based images were rendered in UCSF ChimeraX (v.1.4)[77].

## Isothermal titration calorimetry (ITC) of hGFRα1 interactions with glycosaminoglycans

ITC experiments were performed on a MicroCal PEAQ or MicroCal ITC200 calorimeter (Malvern) (v.1.41) at 20 °C using an assay buffer containing 20 mM HEPES (pH 7.5) and 130 mM NaCl. Protein samples were extensively dialysed against the assay buffer overnight at 4 °C and protein concentrations determined from absorbance at 280 nm measurements. Titrations were performed with 20 µM hGFRα1$^{D1-CT}$ or 60 µM hGFRα1$^{D2-CT}$ in the cell and SOS (200-600 µM) (SantaCruz Biotech) or HS dp10 (65−600 µM) (Iduron) in the syringe. The reference power was set to 5 µcal/s and 20 injections of 4 s injections were recorded for each experiment, with a spacing of 180 s between injections.

Data were analysed using Malvern MicroCal PEAQ-ITC Analysis Software (v.1.41) using non-linear least-squares fitting procedures with the 'One set of sites' model. For each experiment, the heat associated with ligand dilution was measured and subtracted from the raw data. Binding affinities, derived stoichiometries and thermodynamic parameters represent average values from at least 3 independent experiments, with errors quoted as standard deviation from the mean.

## HEK293 suspension cell adhesion assay

HA-rGFRα1$^{154−468}$ (rGFRα1$^{ΔD1}$) was subcloned by restriction digest and ligation, and HA-rGFRα1$^{1−468}$ cis mutants, (rGFRα1$^{FL}$-K251E, rGFRα1$^{FL}$-L287E, rGFRα1$^{FL}$-K251E L287E) were generated by overlap extension PCR mutagenesis of the HA-rGFRα1$^{FL}$ construct[4]. HEK293T cells obtained from the American Tissue Culture Collection (ATCC) were transfected with GFP-expressing plasmid and different rat GFRα1 constructs (ratio of 1:3). Rat GDNF and GFRα1 were used in both this assay and the rat hippocampal neuronal assay. After 48 h the cells were detached with EGTA 1 mM and sorted by FACS to recover GFP$^+$ cells. After centrifugation, the cells were resuspended in DMEM$^+$ 1% SFB, 25 mM HEPES (Invitrogen) at a concentration of 50,000 cells in 200 µl. The adhesion assay was performed in the presence and absence of rat GDNF (rGDNF, 200 ng/ml from R&D Systems). The adhesion assay was performed incubating the cells for 2 h at 37 °C with gentle agitation. After incubation, the cells were plated in multiwells and 10-20 fields/condition were photographed. The percentage of GFP$^+$ cells present in aggregates of more than 5 cells/field were evaluated. For adhesion experiments in the presence of HS, SOS, or human RET$^{ECM}$, cells were preincubated in the presence of the HS dp10 (Iduron) or SOS (SantaCruz Biotech) (0.5 mg/ml) at 37 °C or hRET$^{ECM}$ for 2 h at room temperature with gentle agitation. Then rGDNF (200 ng/ml) was added and the cells were incubated for other 2 h at 37 °C with agitation. Three independent experiments were done for each construct. Statistical significance was calculated using One way ANOVA followed by Tukey´s multiple comparison test. Image analysis was performed in ImageJ (v.1.53a).

## Dendritic spine assay using dissociated hippocampal neurons

Rat hippocampal neurons were isolated from E17.5 Wistar rats as previously described by Ledda et al. 2007[4]. Briefly, rat hippocampal cells from embryonic day E17.5 were obtained by mechanical dissociation of the entire hippocampus and cultured in Neurobasal media (Invitrogen) supplemented with B27 (Invitrogen) supplemented with B27 on 24-well plates with poly-D-Lysine (Sigma-Aldrich)-coated coverslips. Embryos were used independently of their sex. The use of animals was approved by the Animal Care and Use Committee (CICUAL) of the Instituto Leloir according to the Principles for Biomedical Research involving animals of the Council for International Organizations for Medical Sciences and provisions stated on the Guide for the Care and Use of Laboratory Animals.

Transfection was performed at 15 days in vitro (DIV) using Lipofectamine 2000 (Invitrogen) in Neurobasal serum-free medium (Invitrogen) containing 1 µg of total plasmid DNA and 2 µl of lipofectamine per well in 24-well plates. Neurons were co-transfected with the indicated constructs and a plasmid expressing green fluorescent protein (GFP 0.2 µg) and maintained in the presence of rGDNF (150 ng/ml). After 72 h, the cells were fixed with 4% paraformaldehyde PFA followed

by an immunofluorescence of GFP (Invitrogen, dil 1:1000, cat#AB-221569) for the analysis of dendritic spine density. Secondary antibodies were from Jackson ImmunoResearch: Cy2-Donkey anti-Rabbit (cat# 711-225-152, dil 1:200).

For the spine density analysis images were obtained using a Zeiss 710 confocal microscope, using a Plan-APOCHROMAT 63X objective (1.4 NA). Each image corresponds to a merge of 7 optical sections of 0.6 μm each. Neuronal analysis was performed using NeuroJ plugin Image J software. To determine spine density, the number of spines on segments of 100 μm of dendritic length/neuron was counted. Among 14–32 transfected neurons were chosen randomly for quantification experiments. Three independent experiments were performed for each construct. Statistical significance was calculated using One way ANOVA followed by Tukey´s multiple comparison test. Image analysis was performed in ImageJ (v.1.53a).

## Data availability

The refined crystallographic coordinates have been in the RCSB Protein Data Bank (PDB) under accession code 8OS6, the subtomogram averaged map is available in EMDB with the code EMD-18400, and the binned by 4 tomogram of the GDNF-GFRα1 liposome dataset is available in EMDB with the accession code EMD-18651. The source data underlying Figs. 2a, 3a, 4b, 4d, 4f, 5c, 5d, 5e and Supplementary Figs. 1a, 1b, 2a, 2b, 2c, 2g, 3c, 3d, 5a, 5b, 5c, 6a, 6c are provided as a Source Data file. Source data are provided with this paper.

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

## Acknowledgements
We would like to acknowledge the Francis Crick Core flow cytometry and biological research facility STPs. We would also acknowledge the expert help of Annabel Borg (Structural Biology STP) for the expression and purification of recombinant human RET extracellular module and Donald Benton (SB-STP) and Tom Calcraft for assistance with the cryo-ET data collection and processing. N.Q.M. acknowledges that this work was supported by the Francis Crick Institute, which receives its core funding from Cancer Research UK (CC2068), the UK Medical Research Council (CC2068) and the Wellcome Trust (CC2068); N.Q.M. research was also partly funded by the NCI/NIH (grant reference 5R01CA197178). F.L. was supported by Research Career Position from the Argentine Medical Research Council (CONICET). F.L. research was also partly funded by the Argentine Agency for Promotion of Science and Technology (ANPCyT) (grant reference PICT-2019-1472). A.S.R. was supported by fellowships from CONICET and ANPCyT. We would also like to thank Svend Kjær and Professor Mark Lemmon for critical feedback on the manuscript. We thank Mia Zol-Hanlon for critical comments on the manuscript and guidance on proteoglycan analyses. We thank Dr. Chris Earl for providing electron microscopy training. For the purpose of Open Access, the author has applied a CC BY public copyright licence to any Author Accepted Manuscript version arising from this submission.

## Author contributions
F.M.H. & N.Q.M. prepared and wrote the manuscript. S.E.A. carried out the purification, crystallisation, and structure determination of the zebrafish GDNF-GFRα1 complex. F.M.H. carried out the biochemical validation of the zebrafish GDNF-GFRα1 complex, E.M. barrel structure reconstruction, established the liposome adhesion assay and cryo-ET studies. F.M.H. also performed the ITC and SPR experiments. L.M. helped perform ITC experiments and assisted with data analysis. F.L. designed, performed, and analysed cell clustering and synaptogenic assays. A.S.R performed and analysed cell clustering and synaptogenic assays. D.C.B. performed the final zebrafish GDNF-GFRα1 complex structure refinement. A.G.P. assisted with the GDNF-GFRα1 structure determination.

## Funding

## Competing interests
The authors declare no competing interests.
