## [Peer Review File · Nature Communications]

REVIEWER COMMENTS

Reviewer #1 (Remarks to the Author):

This interdisciplinary study describes a beautiful, previously uncharacterised decameric complex formed by the extracellular region of the synaptic adhesion molecule GFR α 1 and its cognate ligand GDNF, and experimental evidence for a functional role of this new super-complex. The authors use both X-ray crystallography and electron microscopy approaches to reveal the intricate arrangement of this large GFR α 1-GDNF complex which forms upon cleavage (possibly also upon sequestration) of the receptor D1 domain. The structural evidence is sound and well presented. The authors also present compelling data suggesting that this assembly contributes to cell-cell adhesion mediated by GFR α 1 and GDNF, using liposome and cell-based assays, and they bring some evidence for a role in synaptogenesis (in vitro). An interesting twist to the story is provided by the structural analysis and biophysical/cell bio experiments that point to a competitive binding mechanism between the decameric complex and binding of other interactors: the RET receptor and HS. This model of competitive binding suggests a mechanism for how GDNF/GFR α 1 could perform different context-dependent functions in vivo. Although definitive proof for a role of the decamer in vivo remains to be shown, the structural and functional data presented here are very convincing and open up unexpected new questions. The paper is incredibly data rich and represents a milestone towards understanding the functions of this important receptor-ligand pair. I recommend publication after consideration of a few points below

Major point:

The data presented indicates that the presence of the GFR α 1 D1 domain inhibits GFR α 1-GDNF decamer formation when using purified proteins, but in HEK aggregation and dendritic spine assays the authors discuss evidence for decamer formation also in presence of the D1 domain. Could this result be discussed in more detail, perhaps even substantiated experimentally, to provide reassurance that the decamer forms in presence of the D1 domain? For example, the hypothesis that NCAM could sequester the D1 domain is tantalising. Does addition of NCAM increase decamer formation in vitro, despite the presence of D1? Or perhaps PLA assays could be used to show that WT proteins 'cluster' more in neurons than the cis-mutants? I appreciate that both of these are challenging experiment, and will be happy with a rephrasing of the discussion to better highlight the limitations of the study, should these experiments not be possible for technical reasons.

Minor points:

- Fig 3b: would it be possible to do sub tomogram averaging to provide a clearer view of the targets please?
- Fig 4e: were postsynaptic markers used to identify dendritic spines? If yes, could this be shown please. If not, the analysis may be biased unless there was another method to identify postsynaptic areas?
- Fig 4b: For the condition where mGFR α 1 Δ D1 is used, the error range is unusually large. More data points could be included to better evaluate the effect of D1 domain removal in this assay.

Reviewer #3 (Remarks to the Author):

Houghton et al. describe the crystal structure, EM pictures and native-PAGE band of a new barrel-shaped 10 GDNF:10 FGFRalpha1 complex from zebrafish. They continue to show that zebrafish GDNF can bridge liposome-bound GFRalpha1 independent of domain 1 of GFRalpha1. Mutation in the GFRalpha1 domain 2 and 3 disrupting GDNF binding (R170E) or cis interaction (K254E and L290E) diminished this liposome interaction as well as a soluble RET fragment. In a HEK cell-based cell adhesion assay the rat cis-interaction mutant GFRalpha1 (K251E and L287E) led to less cell aggregation in the presence of GDNF compared to full-length and domain1 lacking GFRalpha1. The presence of RET reduced cell aggregation in this assay. GDNF induced dendritic spine density in cultured rat hippocampal neurons was higher in cells transfected with full length or domain 1 lacking GFRalpha1 and less after transfection of the cis-interaction mutant of GFRalpha1. Sulphated glycosaminoglycans on domain 1 disrupt the barrel formation but seem not efficient in dissolving it.

I would recommend the following alterations and additions:

Introduction:

Page 3: It would be important to mention that there is cross-reactivity between different GFLs and different GFRalphas (Kramer and Liss, 2015). But there is no binding of GDF15 to GFRalphas and no binding of GFLs to GFRAL which suggests GDF15/GFRAL might be just related TGFbeta family members and not GFL/GFRalpha family members (Saarma and Goldman, 2017).

Page 4: NCAM seems only to bind and signal with GDNF/GFRalpha1 and not other GFL/GFRalpha family members (Paratcha et al., 2003).

Page 4: If mentioning GDNF's neuroprotective action on midbrain dopaminergic neurons it is important to state that GDNF neuroprotective effect depends on the presence of RET and requires therefore the formation of hexameric complexes (Drinkut et al., 2016).

Results:

- These data are mechanistically interesting and would gain further impact if the new barrel-shaped 10 GDNF:10 FGFRalpha1 complex could be confirmed by using native-PAGE or other methods to be present and to influence the cell-based cell adhesion and dendritic spine formation. This is so far not clearly shown.
- The protein expression levels of the GFRalpha1 and RET should be shown for the hippocampal neurons (not only for the HEK293 cells) by Western blotting.
- In the cell-based assays it would be interesting to test the effect of different NCAM and RET forms, and the GFRalpha1 mutant that cannot bind GDNF. For example, soluble NCAM is broadly distributed in the brain and other tissue and it would be of great interest to determine its influence on the barrel-like structure formation. In addition, it would be exciting to know if under physiological conditions cells and tissue form this barrel-shaped 10 GDNF:10 FGFRalpha1 complex and if cells and tissue endogenously expressing RET and/or NCAM have a reduced amount.
- A longer incubation with HS – not only for 2 hours – might be worthwhile to be test on the cell cluster assay to potentially dissolve the complex (SFig. 5).
- Does soluble GFRalpha1 and GDNF also lead to the formation of barrel-like structures or does soluble GFRalpha1 interfere with it?
- What happens in the presence of RET, NCAM, HSPC and soluble GFRa1 in the cellular assays? Are their effects additive or not?
- Does GFRalpha2 with GDNF or Neurturin form a similar barrel-like complex?

Minor points:

Page 12 line 258: mention Fig 3

Page 12 line 262 SFig perhaps 3c instead of 3d

Page 18 SFig 4d shows most likely HEK Western blot, not hippocampal neuron Western blot

Page 19 line 426 introduce the abbreviation HSPG

Page 50 line 1200 Fig legend 4 "or" instead of "of"

The results of this work are noteworthy and significant progress in the field. The description of the work is of sufficient detail and the methodology is sound. The revision should mainly focus on supporting the physiological relevance of the new barrel-shaped 10 GDNF:10 FGFRalpha1. All other points mentioned above can be addressed and discussed in the text.

Point-by-point rebuttal of the reviewers

Reviewer 1:

We were delighted the reviewer recognises our manuscript is “a milestone in understanding GDNF-GFR α 1” and “represents a milestone towards understanding the functions of this important receptor-ligand pair”. The major point raised relates to the role of GFR α 1 D1 domain in preventing assembly formation in vitro but full-length constructs with the D1 domain are fully able to drive adhesion in cells.

We have added a more detailed discussion as suggested by Reviewer 1 on this issue as the full-length protein is substantially more difficult to handle in the absence of adding the GDNF ligand than the Δ D1, precluding liposome binding experiments for full-length GFR α 1. Our best evidence for decamer formation in presence of the D1 domain is from HEK293 cell and hippocampal neurons, where we show that GFR α 1 FL drives adhesion in an GFR α 1 interface EE-mutant dependent manner, implying the same interface multimer is used. Furthermore, we showed these experiments used a full length GFR α 1, and that the D1 domain was not proteolytically clipped by Western blot (Supplementary Fig 5 c).

The points raised by the reviewer on NCAM interaction with the D1 domain are valid but the thrust of this manuscript is to describe the architecture of the decameric complex, its assembly and to report at least two regulatory interactions. Previous literature has reported the NCAM binds to the GFR α 1 D1 domain through its Ig4 domain [PMID : 18353777 see Figure 3 from Sjöstrand & Ibáñez 2008 below] and neurons have high levels of NCAM expression at the synapse that could be interacting with the D1 domain [PMID: 27707798, 17310246]. Therefore in our view, it is reasonable speculation that NCAM interaction with D1 domain links it to the decameric complex. However, it is not easy to demonstrate the link between NCAM and the GDNF-GFR α 1 adhesion complex formation in neurons, as NCAM is involved not only in the adhesion but also in the transduction of the synaptic signalling. However, we have rephrased the discussion as suggested by Reviewer 1 to tone down the discussion around NCAM and GFR α 1.

[Editorial note: this figure was redacted due to third-party rights. It can be found in doi: 10.1074/jbc.M800283200, figures 2b and 3a]

Minor points:

- Fig 3b: would it be possible to do sub tomogram averaging to provide a clearer view of the targets please? We have indeed now included a new figure panel in Figure 3 (and Supplementary Fig. 3) showing results from sub tomogram averaging. These new data clearly show the presence of the decamer with D_5 symmetry in an uncrosslinked state, can be formed by mixing GFR α_1 -anchored liposomes with soluble purified GDNF.
- Fig 4e: were postsynaptic markers used to identify dendritic spines? If yes, could this be shown please. If not, the analysis may be biased unless there was another method to identify postsynaptic areas? We have not used postsynaptic markers, the spines have been recognized morphologically. In previous studies, we have used the same criteria to define spines and validating them with by using postsynaptic markers [PMID: 27707798, see Figure 6 below]

[Editorial note: this figure was redacted due to third-party rights. It can be found in doi: 10.1242/dev.140350, figure 6]

- Fig 4b: For the condition where mGFR α 1 Δ D1 is used, the error range is unusually large. More data points could be included to better evaluate the effect of D1 domain removal in this assay.

A new Figure 4b panel is included showing more experimental data in this analysis lowering the error estimates substantially. With this additional data mGFR α 1 Δ D1 still shows no statistical difference in HEK293 clustering to mGFR α 1FL in the presence of GDNF.

Reviewer 2:

We are glad the reviewer appreciated our “results shown are important for GDNF-mediated synapse formation and adhesion “

Point 1

However, the evidence that these cis-interactions lead to the formation of cis-pentamers (decamers) is weak, as specified in more detail below. Here is the list of issues that should be resolved.

We removed text in discussion about decamer assembly, we cannot state definitively whether in a cellular context assembly follows a pentamer-pentamer bridging or a 2:2 GDNF-GFR α 1 complex multimerization. We discuss these limitations in paragraph 5 of the discussion. However, our cis interface EE mutant impacts on the adhesion level in our liposome and HEK293 clustering assays as well as on the dendritic spine density in dissociated hippocampal neurons. Therefore, we have rephrased in the text that this cis interface is relevant for synaptic-adhesion rather than the mutation explicitly providing evidence for the decamer in these contexts.

1. As the authors are aware that the observed X-ray crystal pentameric structure may be an artefact of crystal packing, they proceeded to electron microscopy where proteins can be imaged at high resolution under conditions that are closer to physiological.

We also show the same decamer by cryo-electron tomography bridging between two liposomes in an un-crosslinked state assembled from individual purified components (Figure 3 C). This strongly argues the decamer is not a crosslinking artefact.

(a) The authors show a single particle average of negatively stained GDNF-GFR α 1 complex that is consistent with the crystal structure. However, the complexes were cross-linked with glutaraldehyde during purification and only 3132 out of 52913 particles were used to obtain the final average. This may indicate that only a small fraction of complexes adopted the pentameric structures and even those might have been induced by cross-linking.

Our new cryo-electron tomography data shows the same decamer assembles bridge between two liposomes in an un-crosslinked state. For the negative stain reconstruction, many 3D classes from picked particles are all discernably the same decamer structure but give an overall lower resolution reconstruction. We only picked the best particle stack to give the best reconstruction.

(b) Furthermore, the authors obtained cryo-electron tomograms of GFR α 1 conjugated to liposomes in the presence of GDNF. They state that the observed densities that interlink liposomes have molecular dimensions consistent with the decameric GDNF-GFR α 1 complexes. However, from 12 tomograms, only three complexes are shown on Fig 2b without any quantitative assessment. It was not shown how their size was measured and there is no scale bar provided. If anything, judged by the membrane thickness, the complexes appear to be less than 10 nm, although they should be around 13.8 nm long (Fig 1b). It is not clear whether another substance is present in the sample that could form the observed structures and possibly also explain the non-spherical shape of liposomes. Therefore, controls are needed to show that the observed liposome linkers are really GDNF-GFR α 1 complexes. Namely, these experiments should be repeated with GFR α 1 mutants where trans and cis interfaces are targeted (as used in other parts of this study) and the data should be quantified.

We have now included a sub-tomogram analysis addressing the size of the assembly and that the observed particles are indeed the decameric assembly we previously showed by crystallography. More examples of particles are shown in Supplementary Fig. 3b and orthogonal views of the 3D tomogram reconstruction are included in Figure 3. A scale bar is added more clearly to Figure 3 – 20 μ m on the bottom RH image size. We have enlarged the scale bar to make it easier for readers. We have re-checked the dimensions, based on a calibrated pixel size which is 3.25 \AA /pixel from IMOD software.

The sub-tomogram average of the decameric complex (two pentameric rings bridged by 5 stalks) confirms that both cis and trans interfaces are relevant for the adhesion complex. These controls were used for the liposome adhesion assay in Figure 3a, and we can confirm that the adhesion driven by GFR α 1D2-D3+ is indeed mediated by the decameric complex from our sub-tomogram averaging experiment.

2. The authors show that clipping of the D1 domain of GFR1 is a slow process that is necessary for the formation of the 700 kDa assemblies, presumably the pentameric complexes (Fig 2a, lines 187-192). Judging by the same panel, it appears that the formation of the pentameric complex requires 10s of hours after clipping of D1, because at the 144 hours lanes D1 is mostly cleaved but the band corresponding to the 700 kDa complex is still weak. This raises the question whether the formation of a 700 kDa complex is physiologically relevant.

The clipping is slow in vitro as it arises from a contaminating protease impurity. We have no evidence that a protease clips in cellulo as we see only full length GFRa1 by Western blots. We have therefore moved the clipping experiment to Supplementary Figure 2. We have revised the text in the results section to remove the emphasis of the D1 clipping in vitro and instead focused on the decameric complex forming from the zGFRa1D2-D3+ construct.

3. (Lines 421-425, 492-494 and elsewhere) The crystal structure of GDNF-GFR1 (13.8 nm) is much shorter than the synaptic cleft with (estimated to 20 nm by electron microscopy of resin-embedded samples and around 25 nm in cryo-preserved samples). consequently, it is not clear how it could serve as an trans-synaptic complex.

This complex lacks another 65aa from the C-terminal of each GFRa1 protomer that would be present in the full length and would compensate allowing GFRa1 to bridge across the synaptic cleft gap. We have highlighted how the C-tails of GFRa1 may couple the decamer to two opposing membranes across the cleft in a schematic in Supplementary Fig 4a.

4. The average structure obtained by the single particle analysis should be deposited in a public database.

We have deposited the cryo-electron tomography reconstruction as it is un-crosslinked into a public EM/ET database (Accession code EMD18400).

5. Fourier shell correlation for the single particle average should be shown.

We have included for the NS-EM map in Supplementary Fig 2f and for the cryo-ET map in Supplementary Fig.3d.

6. At least one cryo-electron tomogram should be made available for reviewers. It would be useful to also deposit one or more tomograms to a public database.

We have included a movie of one of our tomograms linking it as a Supplementary Video.

7. (Lines 721-723) The units "d/nm" are not clear. It would be useful to show a distribution of liposome hydrodynamic radii obtained by dynamic light scattering.

We refer to "d/nm" as a standard unit diameter of hydrodynamic radius. We have added a distribution of liposome hydrodynamic radii in Supplementary Fig 3a and clarified in figure legend the unit d.nm = particle diameter in nm.

8. (Lines 727-730) It is not clear to me how was absorption at 650 nm used to determine the adhesion of liposomes. Perhaps the authors meant dynamic light scattering that they used to assess the size distribution of liposomes. If that is the case, this should be also corrected in results

We measured turbidity/optical density at 650nm which is an indirect measure of lipid aggregation as used by Harrison et al 2011 [PMID: 21300292]. Dynamic light scattering was only used to check the size distribution of liposomes following extrusion, serving as a preparation quality control shown in Supplementary Fig 3a.

9. (Lines 199-201 and elsewhere) The authors state that the GDNF-GFR1D2-D3+ is a minimal portion that is necessary and sufficient to drive the complex formation. They show that this portion is sufficient, which is an important result. However, because smaller portions of GFR1 were not tested, it is not clear that the GDNF-GFR1D2-D3+portion is the smallest or necessary.

We thank the reviewer for pointing this out and we have adjusted the text accordingly.

10. (Lines 306-309) In order to show that zGDNFmat shifted from soluble to membrane-bound fractions when incubated with GFR1D2-D3+, it is necessary to show P and S lanes when zGDNFmat is present and GFR1D2-D3+ is absent.

GDNF alone does not bind non-specifically to Ni-NTA DGS lipids without GFRa1 loaded onto the liposomes. This is removed when all Ni-NTA DGS lipids are saturated with His-tagged GFRa1 (ie the conditions used in these experiments). Therefore the more appropriate negative control we used was R170E mutant that no longer binds GDNF. With zGFRa1^{D2-D3+}-R170E-coated liposomes GDNF is predominantly in the soluble fraction (Supplementary Fig 4c), whereas for zGFRa1^{D2-D3+}-coated liposomes and zGFRa1^{D2-D3+}-K254E L290E-coated liposomes, GDNF partitions into the pellet fraction. This demonstrates specific binding of GDNF to membrane-conjugated GFRa1.

11. (Lines 321-324) Concentrations of Ca²⁺, EDTA, HS and the proteins should be stated.

We thank the reviewer for pointing this out and we have added this information to the M&Ms describing a native-PAGE analysis of zGDNF-zGFRa1 complexes.

12. (Lines 386-388) To be precise, the observation that GDNF cell adhesion is unchanged upon D1 deletion shows that the GFR1D1 is sufficient, but this alone does not mean it is necessary.

We thank the reviewer for pointing this out and we have adjusted the text accordingly.

13. (Lines 391-392) The formulation that in vitro adhesion is "independent of the GFR1 D1 domain" is misleading, because the authors showed that D1 prohibits in vitro adhesion, so "precluded by" would be better than "independent of".

We thank the reviewer for pointing this out. We have adjusted the text to indicate that the adhesion complex forms in the absence of the D1 domain (mGFRa1ΔD1) in our HEK293 clustering assay, whereas the D1 domain precludes the formation of the complex in solution.

14. (Lines 392-394) Another explanation is that the pentamer is not necessary for adhesion in cellulo. The observation that in vitro cis dimers or trimers exists in the presence of D1 (as evidenced by the presence of bands heavier than 242 kDa at all times on Fig 2a) supports this possibility.

We think for the mammalian complex, the interface is relevant and this is where we put emphasis in the discussion as we cannot directly show the decamer forms in a human GDNF-GFRa1 context. We have toned down any claims that the human system is shown to form decamers but it does indeed use the cis interface.

15. (Lines 494-498) The authors clearly state that RET and HS promote GFR1 conformational change. Therefore, in my opinion, stating that GFR1 is an active conformational sensor is not adequate.

We thank the reviewer for this suggestion. We therefore toned down and changed "active" to "conformational plasticity" in GFRa1 dictates whether it signals through an adhesion complex or a trophic support complex.

16. (Lines 512-516, Figure 6) The authors propose that the role of RET in disturbing the GDNF-GFR1 complexes relevant for the maturation of newly formed synapses. However, RET has been reported not to be present in the hippocampus and forebrain (ref 45). This issue should be discussed.

We have added a section at the end of paragraph 2 of the discussion to address this – hippocampal neurons indeed express no RET and therefore are able to form adhesive interactions through GDNF-GFRa1, whereas dopaminergic, motor and enteric neurons express RET and GFRa1 so promote trophic signalling. However, a third potential context is where RET expression is induced in neurons transiently, for example by calcium release, and in this context there may be a switch from adhesion to trophic support.

17. (Line 258-9) Figure 3 should be cited.

Figure 3 now cited correctly.

18. (Line 286) Citation is needed.

Citations added

19. (Line 446) The definition of dp10 should be moved where it is mentioned for the first time.

Definition moved

20. (Line 727) "then" instead of "them".

Done

21. The readability would be improved by streamlining the nomenclature. For example, "heparan sulfate proteoglycans", "heparan sulfate", HS, HSPG and GAG all mean the same thing. Also, using LiCAM to mean specifically GFR₁ (line 522) is not optimal.

We have changed more instances to using HS but we have kept GAG where we refer to both HS and SOS.

22. (Lines 797-799) It would be useful to briefly summarize the neuronal cultures protocol of Ref 4 and to cite the primary reference for that protocol.

We have added a short summary in M&Ms from Ledda et al 2007 [PMID: 17310246] including how the neurons were cultured.

23. (Figure 2d, e and 3b) Scale bars are missing.

Enlarged scale bars have been added to improve clarity

24. (Figure 3a) The black and the blue symbols in the legend are hard to distinguish.

Changed colours to brighter blue for better contrast as suggested by the reviewer.

Reviewer #3 (Remarks to the Author):

We thank reviewer 3 for recognising "The results of this work are noteworthy and significant progress in the field. The description of the work is of sufficient detail and the methodology is sound."

Introduction:

Page 3: It would be important to mention that there is cross-reactivity between different GFLs and different GFRalphas (Kramer and Liss, 2015). But there is no binding of GDF15 to GFRalphas and no binding of GFLs to GFRAL which suggests GDF15/GFRAL might be just related TGFbeta family members and not GFL/GFRalpha family members (Saarma and Goldman, 2017).

We thank the reviewer for this suggestion. We have adjusted the introduction text accordingly to state GDF15 is distant relative of GFLs.

Page 4: NCAM seems only to bind and signal with GDNF/GFRalpha1 and not other GFL/GFRalpha family members (Paratcha et al., 2003).

We thank the reviewer for noticing this, we adjusted the text accordingly.

Page 4: If mentioning GDNF's neuroprotective action on midbrain dopaminergic neurons it is important to state that GDNF neuroprotective effect depends on the presence of RET and requires therefore the formation of hexameric complexes (Drinkut et al., 2016).

We recognise this is an important point so we have added a comment, "Therapeutic interest in GDNF has stemmed from its known neuroprotective action on midbrain dopaminergic neurons both *in vitro* and *in vivo*²⁴⁻²⁷. This neurotrophic behaviour is driven by engagement and activation of RET by the GDNF-GFRα1 2:2 complex²⁸⁻³¹."

Results:

- These data are mechanistically interesting and would gain further impact if the new barrel-shaped 10 GDNF:10 FGFRα1 complex could be confirmed by using native-PAGE or other methods to be present and to influence the cell-based cell adhesion and dendritic spine formation. This is so far not clearly shown.

We have indeed showed this by native PAGE on Figure 2 and also have included in Figure 3 a full reconstruction of the decameric complex bridging across adhering liposome membranes. This sub-tomogram averaging reconstruction using un-crosslinked sample confirms the complex forms stable assembly on liposomes. This combines with our cell-based adhesion assay and dendritic spine assay (Fig. 4) to demonstrate that mutants targeting the pentameric interface impact cell clustering and synaptogenesis.

- The protein expression levels of the GFRα1 and RET should be shown for the hippocampal neurons (not only for the HEK293 cells) by Western blotting.

We thank the reviewer for this suggestion. We have included a figure showing the expression levels of GFRα1 mutants in hippocampal neurons in Supplementary Fig 5d. Hippocampal neurons express endogenous levels of GFRα1 and NCAM but do not express Ret. These has been previously demonstrated in (i) Irala et al 2016 [PMID: 27707798] (Development, Figure 1A and B), (ii) Bonafina et al 2019 [PMID: 31875542](Cell Reports, Figure S1) and (iii) Ledda et al 2007 [PMID: 17310246] (Nat. Neuroscience, Figure 1)

- In the cell-based assays it would be interesting to test the effect of different NCAM and RET forms, and the GFRα1 mutant that cannot bind GDNF. For example, soluble NCAM is broadly distributed in the brain and other tissue and it would be of great interest to determine its influence on the barrel-like structure formation. In addition, it would be exciting to know if under physiological conditions cells and tissue form this barrel-shaped 10 GDNF:10 FGFRα1 complex and if cells and tissue endogenously expressing RET and/or NCAM have a reduced amount.

This are excellent ideas but we feel they are beyond the scope of this already full manuscript. We tested the GFRα1 mutant in synaptogenesis assay. Ledda et al 2007 and Irala et al. 2016 [PMID: 27707798, 17310246] have previously used siRNA against NCAM in hippocampal neurons showing reduced level of dendritic spines and presynaptic differentiation markers induced by GDNF when NCAM expression is lost. The reviewer asks if the decamer complex forms *in vivo* with NCAM, we currently lack imaging tools to show this and we have adapted the discussion to reflect that other unknown partner proteins for GFRα1 may sequester the D1 domain.

Regarding full-length RET at the cell surface, we found that cell-surface full-length RET impacts the expression of GFRα1 (see figure below). This was why we decided to do the experiment in the presence of soluble extracellular RET.

(A) Effect of cell-surface RET expression on GDNF-induced HEK293 cell adhesion assay. The level of cell adhesion promoted by transfected proteins with and without GDNF treatment evaluated as the percentage of GFP+ cells present in aggregates more than 5 cells/field \pm s.e.m. from at least three independent experiments. ANOVA followed by Tukey's multiple comparisons test. * $p < 0.05$, ** $p < 0.01$, *** $p < 0.001$. **(B)** Western blot to show the impact of full-length RET on expression levels of mGFR α 1 in HEK293 cells. mRET expression interferes with the expression of mature mGFR α 1. mGFR α 1 detected using an anti-HA and RET detected using a polyclonal anti-RET antibody.

- A longer incubation with HS – not only for 2 hours – might be worthwhile to be test on the cell cluster assay to potentially dissolve the complex (SFig. 5).

We have followed previous described protocols for cell clustering with exogenous GAGs (Seoung Youn Won et al 2017 [PMID: 29081732] *Frontiers in Molecular Neuroscience*, Figure 5). Extending the period of incubation for the adhesion, we were concerned this could potentially affect cell viability. In our reported experiments the cells have already been pre-incubated for 2 h in the presence of HS and then incubated a further 2 h with the addition of GDNF.

- Does soluble GFR α 1 and GDNF also lead to the formation of barrel-like structures or does soluble GFR α 1 interfere with it?

The reviewer raises an interesting point. In principle it should be possible to form a barrel structure in solution (given the visualisation by negative stain EM) and as GFR α 1 is known to be shedded in vivo enabling signalling with GDNF at more remote sites. In previous studies (Ledda et al 2007 [PMID: 17310246], see their Figure 2f) we have demonstrated that the addition of soluble GFR α 1 blocked GDNF-induced adhesion of GFR1-expressing cells, whereas soluble GFR α 3, which does not bind GDNF, had no effect.

- What happens in the presence of RET, NCAM, HSPC and soluble GFR α 1 in the cellular assays? Are their effects additive or not?

We have not tested if all of these effect are additive. This could be an interesting experiment in the future but we think it is beyond the scope of this study.

- Does GFR α 2 with GDNF or Neurturin form a similar barrel-like complex?

This is a good point but we have not tested this, though the interfaces are quite well conserved and the 2:2 complexes and angle fit could equally well for neurutrin-GFRa2 complexes into the negative stain reconstruction.

Minor points:

Page 12 lane 258: mention Fig 3

Corrected

Page 12 lane 262 SFig perhaps 3c instead of 3d

Corrected

Page 18 SFig 4d shows most likely HEK Western blot, not hippocampal neuron Western blot

Correct, we thank the reviewer for spotting this error. We have added immunofluorescence imaging of hippocampal neurons to Supplementary Fig 5d.

Page 19 lane 426 introduce the abbreviation HSPG

We have introduced this

Page 50 lane 1200 Fig legend 4 "or" instead of "of"

Corrected

REVIEWERS' COMMENTS

Reviewer #1 (Remarks to the Author):

The authors have addressed my concerns adequately. Congratulations on the excellent work.

Reviewer #2 (Remarks to the Author):

The authors adequately addresses almost all of my points. There are only two outstanding minor issues (numbered according to points I made in the first round).

3. The answer the authors provided to this point should be included in the manuscript. In addition, it should be stated that in the schematic shown in Supplementary Fig 4ai, the distance between membranes corresponds to 15 nm, as estimated from the reported 13.8 nm height of the GDNF-GFR1 decamer crystal structure, which is in contrast to the width of the synaptic cleft reported to be 20 nm by electron microscopy of resin-embedded samples and around 25 nm in cryo-preserved samples.

6. I still recommend that at least one tomogram should be deposited to a public database, such as the Electron Microscopy Data Bank.

Reviewer #3 (Remarks to the Author):

The revision of the manuscript by Houghton et al. has significantly improved the story and made it suitable for publication.

A few typos should still be corrected, for example, line 63 "GDF15" instead of "GDF-15", reference 78 should say "Schrödinger" instead of "Schrodinger", reference 79 should say "Schägger" instead of "Schaägger"...

Point-by-point response to the reviewers' comments

Reviewer #1:

The authors have addressed my concerns adequately. Congratulations on the excellent work. – Thank you!

Reviewer #2:

There are only two outstanding minor issues (numbered according to points I made in the first round).

3. The answer the authors provided to this point should be included in the manuscript. In addition, it should be stated that in the schematic shown in Supplementary Fig 4ai, the distance between membranes corresponds to 15 nm, as estimated from the reported 13.8 nm height of the GDNF-GFR1 decamer crystal structure, which is in contrast to the width of the synaptic cleft reported to be 20 nm by electron microscopy of resin-embedded samples and around 25 nm in cryo-preserved samples. – We have added a comment mentioning this point in the figure legend of Supplementary Fig. 4a and in the main text on page 15.

6. I still recommend that at least one tomogram should be deposited to a public database, such as the Electron Microscopy Data Bank. – We have submitted a reconstructed tomogram, linked to the sub-tomogram average map deposition, which is available in EMDB with the code EMD-18651 as requested.

Reviewer #3:

A few typos should still be corrected, for example, line 63 "GDF15" instead of "GDF-15", reference 78 should say "Schrödinger" instead of "Schrodinger", reference 79 should say "Schägger" instead of "Schaägger"... – we have corrected these typos as requested